# Unveiling hidden energy poverty using the energy equity gap

Shuchen Cong [1], Destenie Nock [1,2✉], Yueming Lucy Qiu [3✉] & Bo Xing[4]

Income-based energy poverty metrics ignore people's behavior patterns, particularly reducing energy consumption to limit financial stress. We investigate energy-limiting behavior in low-income households using a residential electricity consumption dataset. We first determine the outdoor temperature at which households start using cooling systems, the inflection temperature. Our relative energy poverty metric, the *energy equity gap*, is defined as the difference in the inflection temperatures between low and high-income groups. In our study region, we estimate the energy equity gap to be between 4.7–7.5 °F (2.6–4.2 °C). Within a sample of 4577 households, we found 86 energy-poor and 214 energy-insecure households. In contrast, the income-based energy poverty metric, energy burden (10% threshold), identified 141 households as energy-insecure. Only three households overlap between our energy equity gap and the income-based measure. Thus, the energy equity gap reveals a hidden but complementary aspect of energy poverty and insecurity.

[1] Department of Engineering and Public Policy, Carnegie Mellon University, Pittsburg, PA, USA. [2] Department of Civil and Environmental Engineering, Carnegie Mellon University, Pittsburg, PA, USA. [3] School of Public Policy, University of Maryland College Park, College Park, MD, USA. [4] Department of Forecasting, Resource Planning and Development, Salt River Project, Tempe, AZ, USA. ✉email: dnock@andrew.cmu.edu; yqiu16@umd.edu

Energy poverty manifests itself in a high percentage of income spent covering energy bills, increased risk of electricity shutoffs, and a household's inability to maintain comfortable indoor temperatures or use desired services (e.g., air conditioning, heat, computers)[1,2]. An often-overlooked space in energy poverty analysis lies in the cavity between metrics that measure financial stress (energy burden defined as energy expenditure over total income) and complete lack of energy services (utility shutoffs). Within this cavity are the households which limit their energy consumption to reduce financial strain. These households may appear to spend small amounts of their income on their energy bills while limiting enough energy to avoid having the utility cut their power supply. It is estimated that, annually, 1300 people die every year in the U.S. from extreme heat[3]. In 2009 and 2010 alone, over 8250 emergency room visits in the US were caused by heat stroke[4], with low-income, minority, and elderly populations being disproportionally affected[3]. A large portion of these deaths may have been prevented if people could cool their homes properly. We acknowledge that proper cooling ability includes being able to acquire and sufficiently use an AC system[5,6].

As the effects of climate change manifest themselves in heatwaves[7] and deep freezes, communities will need to adapt (i.e., reduce their risk of illness and death[8]) by creating comfortable indoor temperatures within their homes. However, this depends on whether they can rely on their resources for adopting energy-efficient heating and cooling systems, meaning many vulnerable households who limit their energy consumption, potentially putting themselves at risk of heatstroke or hypothermia, may not qualify for energy poverty alleviation under current programs.

For example, in the US, the two main energy assistance programs, the Low Income Home Energy Assistance Program (LIHEAP) and the Weatherization Assistance Program (WAP), use an income threshold to determine eligibility[9,10]. LIHEAP uses reduction in a household's energy burden, service loss prevention, and service restoration, to calculate the effectiveness of its assistance[9]. WAP is geared more towards homeowners with its performance based on the number of households weatherized and post-weatherization surveys for those receiving assistance, with questions including change in energy burden and change in forgoing other necessities like food to pay energy bills[9,11]. This suggests that these programs make an implicit assumption that people meet or try to meet their energy needs first compared to other necessities like food or healthcare. Neither WAP nor LIHEAP takes into explicit account those who forgo energy consumption to pay for other necessities (i.e., energy limiting behavior), nor do they offer a clear definition of energy poverty[9]. The lack of consideration for households who forgo energy for other needs poses a limitation in identifying the multidimensional nature of poverty[12,13] and reduces the options for policy intervention. In LIHEAP and WAP, spending patterns are viewed as adequate ways to measure the effectiveness of these energy assistance programs[9,11].

In a broader sense, energy poverty is defined as insufficient energy access due to lack of supply, low affordability, limited quantity, poor quality, unreliability, or a combination of these shortcomings. Existing energy poverty metrics fall into the following categories: A) primary or secondary, and B) relative or absolute, as seen in Fig. 1. Here we define each category combination and provide some examples of each. A primary metric is defined as a metric that directly utilizes consumer-level information. A secondary metric would require derivation to reach a conclusion. Secondary metrics include metrics that aggregate utility information or use weighted scoring for poverty indices. A relative metric compares the energy poverty status of two or more entities (i.e., country-to-country or household-to-household) or

one with oneself (i.e., progress over time for one country). Finally, an absolute metric will provide a strict threshold for energy poverty.

Relative-secondary metrics for energy poverty use summary statistics from the regional or local level (not individuals) and compare the progress of different regions to some benchmark. These are often used to describe progress in energy poverty reduction in developing countries. One example is an access-consumption matrix at a national level[14]. Access-consumption matrices portray shifts in a country's energy profile, mainly the change in fuel utilization and how many people use each fuel. If more people are gaining access to energy services in an underdeveloped country, and more people are shifting from dirtier to cleaner fuels in developing regions, energy poverty is reported as decreasing[14,15]. Due to the fact that energy poverty in developing regions often means a lack of access to modern energy services, these national energy access metrics are best used for countries beginning their clean energy transition and expanding access to modern energy services[14,15].

Relative-primary metrics come directly from households or individuals, and benchmark feelings of energy poverty in comparison to others in the population. These can be scores from a survey asking questions on self-perception of energy poverty. For example, a survey done in Greece used indicators such as "inability to keep home adequately warm," "leakages, damp walls, mold," and "restriction of other essential needs" to solicit the subjective feeling of energy poverty[16,17]. Compared with the 10% energy burden threshold, the study found that when a household is objectively categorized as energy-poor, they were more likely to respond "yes" for the subjective indicators. Another study explores the relationship between social relations and energy access, where a positive feedback loop exists between good social relations and higher quality energy access[2,18]. However, drawbacks of survey-based metrics are long completion times and difficulty comparing the level of energy poverty experienced

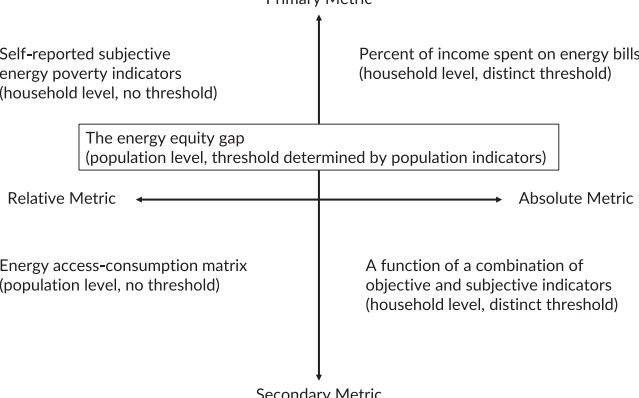

**Fig. 1 Categories of energy poverty metrics.** The *X*-axis represents relative or absolute metrics, or whether the metric has a distinct threshold for energy poverty. The *Y*-axis represents primary or secondary metrics, or whether a metric requires more than basic consumer-level data to calculate. A primary-absolute metric can be energy burden, or percent of income spent on energy bills; a primary-relative metric can be self-reported energy poverty indicators; a secondary-absolute metric can be a combination of the previous two, where an arbitrary score is calculated based on energy burden and survey results; and a secondary-relative metric can be an energy access-consumption matrix, often used to portray the progress of energy access in developing countries. The energy equity gap is a primary metric that can be both relative and absolute, where it can tell us the relative energy equity progress of a region, as well as household-level energy poverty.

between households. Although there are many survey studies on housing characteristics and perceived energy poverty[16,19,20], we find a lack of metrics which can quantify the amount of energy a household forgoes to alleviate financial burden (i.e., energy limiting behavior). Thus, there is a need to use surveys in combination with data driven approaches to elicit perceptions of energy limited behavior, while also determining the degree to which people actually limit energy consumption.

Absolute-secondary metrics compare information related to energy usage against some predetermined threshold. These metrics often combine income-based metrics with socio-demographic factors and housing conditions to calculate a weighted score to measure energy poverty[2,18]. However, because these measures (income-based, survey, and combined) single out those currently experiencing energy poverty, they can miss distributional changes over time and the severity of the energy poverty experience relative to the rest of the population. Additionally, a limitation is that these metrics often focus on equality (i.e., all households reaching a certain status) but miss equity because they cannot identify how the energy-poor compare to the non-energy-poor. Achieving energy equality entails giving everyone the same tools to achieve a desired level of energy consumption. For example, providing households with a voucher to spend less than 10% of their income on their current energy bills. Achieving energy equity entails giving groups different types of tools such that they can equally take advantage of opportunities or reach a desired goal. For example, an energy equity policy could entail each person receiving enough assistance to cool or heat their homes to their desired comfortable temperature.

Absolute-primary metrics use individual or household information and measure energy usage and other information against some predetermined threshold. These metrics are often used to measure household-level energy burden (i.e., energy expenditure over income). The underlying theory is that the more significant percentage of income spent on energy, the more energy-poor one is, similar to Engel's Coefficient for food expenditure[21]. A common threshold for energy burden to indicate a household is energy poor (proposed in 1991) is 10%[9,18]. Energy burden can depend on a number of factors such as electricity price, prices of other goods, and heating or cooling needs[1]. The advantage of this threshold metric is that it indicates the economic burden of meeting energy needs and does not have a high computational burden. However, limitations of using the energy burden threshold metric include not distinguishing between gross (i.e., pre-tax) and disposable income (i.e., post-tax and other mandatory charges like mortgage and rent), not considering indoor comfort levels, and not considering local and current costs of living[18]. As a result, households may have the same pre-tax income level but have vastly different mortgage or rental costs, meaning that the 10% threshold on gross income may miss people who are spending more than 90% of their income on other basic necessities[22].

Despite the recent development of new metrics to capture consumer behaviors (e.g., under-consumption of energy and choice of thermal comfort)[3,8], household-level energy poverty evaluation for government assistance programs in developed countries has been led by absolute-primary income-based metrics[9,18,23,24]. While these income-based metrics are widely used, they have a few shortcomings. First, they are sensitive to energy prices and mask the degree to which households change their energy consumption behavior following a price shift[18]. Second, and perhaps more importantly, the energy burden metric does not capture vulnerable low-income households who forgo energy usage to reduce financial stress. Lastly, income-based metrics miss essential dimensions of energy poverty, such as the inability to use enough energy to cool or heat homes to comfortable, and/or safe temperatures[25].

Most papers investigating the inability to satisfy a household's desired energy demand focus on electricity supply and reliability constraints in developing countries[26–28] and energy affordability in developed countries[1,29,30]. Within developed countries (i.e., those with close to 100% electricity supply access)[31], energy poverty and insecurity can manifest themselves in 1) electricity shutoffs resulting from nonpayment, 2) forgoing heating services due to financial strain and participating in unsafe practices (e.g., using the stove or oven for heat, unsafe uses of space heating technologies which lead to fires[32]), 3) spending a large percentage of income on energy bills, and 4) difficulty adopting clean energy and efficient technologies[17,33–35]. While multiple papers address the indicators of energy poverty[9,36,37] and insecurity[17,35], we find a void of metrics that can identify energy-limiting households (i.e., those without comfortable indoor temperatures) who may put themselves at risk of heat-related illness, excess indoor moisture, mold growth, and other adverse health effects[35,38,39] (e.g., respiratory illness and asthma).

In countries where the entire population has access to modern energy infrastructure, household-level energy poverty manifests itself as having inadequate energy services within the household, or an inability to consume energy at a desired level[2]. Thus, a more holistic definition of energy poverty would include people who limit their energy consumption (i.e., display energy limiting behavior), and those who spend a large portion of their income on their energy bills (i.e., high energy burden). Our work fills this gap by creating a metric which can identify energy limiting behavior. We define energy limiting behavior as a household's inability or unwillingness to consume enough energy to reach a desired level of comfort. A household displays energy limiting behavior if they reduce their energy consumption significantly below another household within the same region that does not have a budget constraint for energy spending. For example, assume households A and B live in the same region and have similar preferences for their ideal indoor temperature, around 70 °F (21 °C). Household A is a low-income household (i.e., a budget constraint on energy spending), and Household B is a high-income household (i.e., no budget constraint). If Household B starts using their air conditioning unit when it is 70 °F (21 °C) outside, but Household A waits until it is 75 °F (24 °C) outside, then household A is displaying 5 °F (3 °C) of energy limiting behavior compared to Household B.

Here, we introduce a behavior-based energy poverty measure, the energy equity gap, which captures one critical aspect of energy poverty (i.e., energy-limiting behavior), providing a complementary metric to capture inequity within a region. We first determine the outdoor temperature at which households start using cooling systems, the inflection temperature. Then, our relative energy poverty metric, the energy equity gap, is defined as the difference in the inflection temperatures between low and high-income groups. In our study region, we estimate the energy equity gap to be between 4.7 °F (2.6 °C) and 7.5 °F (4.2 °C). In 2015–2016, within our sample of 4577 households, we found 86 energy-poor and 214 energy-insecure households. In contrast, the income-based energy poverty metric, energy burden, identified 141 households as energy insecure when the threshold is set to 10%, with only three households overlapping between our energy equity gap and the income-based measure.

## Results

**Quantifying residential electricity consumption patterns.** To capture those households left behind by income-based energy poverty measures, we propose a different energy poverty metric: the energy equity gap. We illustrate its effectiveness for identifying households at risk for inability to reach comfortable indoor

temperatures, and possibly heat-related illness. Our study region in the US, Arizona, has long, high-heat summers and mild winters (see Supplementary Information Note 6). Arizona has a higher level of heat-related illnesses (2944 heat-related ER visits in 2019[40]) compared to cold-related illnesses (495 cold-related ER visits in 2019[41]), leading us to focus our energy poverty analysis on the electricity sector due to this providing the bulk of cooling energy in the summer (air conditioning (AC) or fan usage). Identifying cooling system use is vital for addressing and planning for energy justice, which hinges on the proper distribution of benefits for a clean energy transition[42–44] and an ability to mitigate the effect of heatwaves. We also introduce a tiered system for identifying and addressing the energy poverty needs of the most vulnerable households and contrast this with the existing income-based metric.

The energy equity gap is a measure that investigates how consumer electricity consumption behavior across income groups shifts with temperature (i.e., consumers' temperature response functions). Previous research has investigated how consumers' temperature response functions (modeling energy usage against temperature or other climate factors) change with climate[45,46], but have not incorporated these functions into energy poverty identification. The energy equity gap metric considers the effect of outdoor temperature on energy consumption and quantifies relative energy limiting behavior, where those with fewer constraints on their budget set the threshold for a desired level of energy consumption to maintain a comfortable indoor temperature in the region (see Methods). Using the energy equity gap, we measure electricity usage patterns between income groups within a metropolitan region, thus eliminating the effect of weather or outdoor temperature on electricity usage for different households, which might occur in large study areas. A benefit of our primary-relative energy poverty metric, energy equity gap, is that policymakers can have more targeted energy justice efforts by first identifying the outdoor temperature that places their region at risk for heat-related illnesses or energy limiting behavior. Once the threshold has been set, policymakers can then use our relative metric to identify energy insecure households that are dangerously close to sinking into energy poverty and create proactive measures for reducing their burden and increasing their ability to consume energy to increase their comfort. In addition to capturing household-level electricity consumption behavior, the energy equity gap allows for a cross-temporal comparison of population-level energy equity within a region.

The basis of the energy equity gap is household-level inflection temperatures. To best incorporate behavior into the metric, we define the inflection temperature as the outdoor temperature at which a household starts using its cooling system as it shifts from spring to summer temperatures, assuming there is no difference in comfort preference or need across income groups. To find the inflection temperature of each household, daily electricity consumption is modeled using average daily temperature, electricity pricing plan, holiday effects, and day-of-the-week and month-of-the-year fixed effects (see Methods). The minimum of the quadratic equation between electricity consumption and temperature after controlling for the covariates mentioned above is defined as the temperature at which people start using their cooling systems, the inflection temperature (Fig. 2). This assumption stems from 1) heating and AC systems being the largest energy consumer within a household[45], and 2) our study region having a warm and dry climate, with short, mild winters and long, high-heat summers. If the study region is in a colder climate or a climate with more distinct seasons, we recommend separating the year into two climate zones (i.e., spring-summer-fall and fall-winter-spring). To adapt the energy equity gap to identify heating system energy use, we would need to include

information from the gas and oil sector. We leave the heating sector analysis for future work. We hypothesize that low-income households are more likely to endure higher temperatures before they start cooling their homes in the summer to save money and will thus have higher inflection temperatures.

**Redefining energy poverty and energy insecurity.** The energy equity gap is defined as the difference between the highest and lowest median household inflection temperatures among all income groups (Fig. 3) of the study region, a metropolitan area in Arizona (see Methods). We chose to use the median instead of the mean to desensitize the measure from outliers. The energy equity gap indicates the disparity in energy use across the income spectrum for a region while eliminating the effect of climate and electricity pricing. Within our metropolitan region, we assume the climate is uniform for households in the sample data, and everyone has access to the same energy services. Therefore, a sign of reduced energy inequity would be a narrowing energy equity gap, indicating that households are converging to a similar inflection temperature, thus reduced energy inequity.

The distribution of household-level inflection temperatures across income groups is shown in Fig. 3. We see that the energy equity gap ranges from 4.7–7.5 °F (2.6–4.2 °C), highlighting that low-income groups are more likely to forgo cooling services until later in the summer than high-income groups (see Supplementary Info Note 5 for another sample analysis). Furthermore, there is a statistically significant difference between the median inflection temperatures between each income group for all years, meaning there is little chance that the inflection temperature differences occurred by chance, verified using the Mood's Median test (see Methods).

Figure 4 illustrates the change in the energy equity gap across income groups for the four years in our analysis. The higher inflection temperatures further show low-income households tend to wait longer to turn on their AC units, pointing to underlying constraints, budget or otherwise, restricting their access to cooling. When cooling is restricted, it has been shown that buildings are at a higher risk for increased rates of mold, allergens, and fungi growth[47,48] and that when the degree of

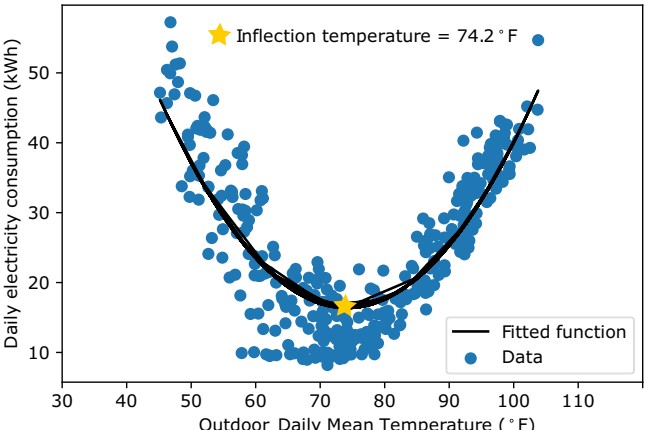

**Fig. 2 Identifying the inflection temperature for daily electricity consumption and local daily mean outdoor temperature.** This graph represents the daily electricity consumption of one household for one year ($N = 365$). The star marks the inflection temperature for this household for this year. We note that our true temperature response function includes electricity price, weekend, holiday, day of the week, and month of the year effects. The inflection temperature is the minimum of the quadratic temperature response function between the residuals after controlling for these factors and outdoor temperature.

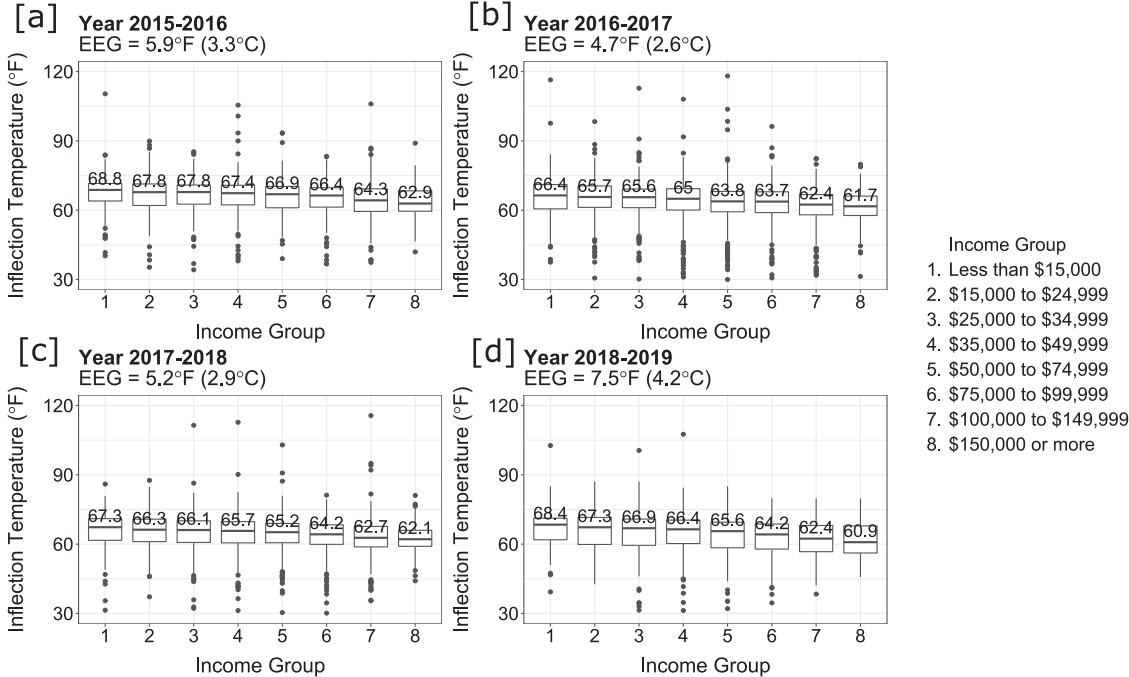

**Fig. 3 The distribution of inflection temperature across income groups.** The energy equity gap (EEG) for each year is calculated as the difference between the highest and lowest median inflection temperature (indicated by the middle bar and number) among all income groups in all four panels, income group 1 had the highest, and income group 8 had the lowest median inflection temperature. The energy equity gap (EEG) is shown at the top of each panel. (**a**) 2015–2016 N = 4577 households, (**b**) 2016–2017 N = 4522 households, (**c**) 2017–2018 N = 3852 households, (**d**) 2018–2019 N = 2650 households. Each box and whiskers plot indicates the minima and maxima of inflection temperatures of one income group for one year (the lower and upper bound of the whiskers), the first and third quantiles (the lower and upper bound of the box), and the median (the middle line). The outliers are shown as dots on either side of the whiskers. Source data can be found in our code repository.

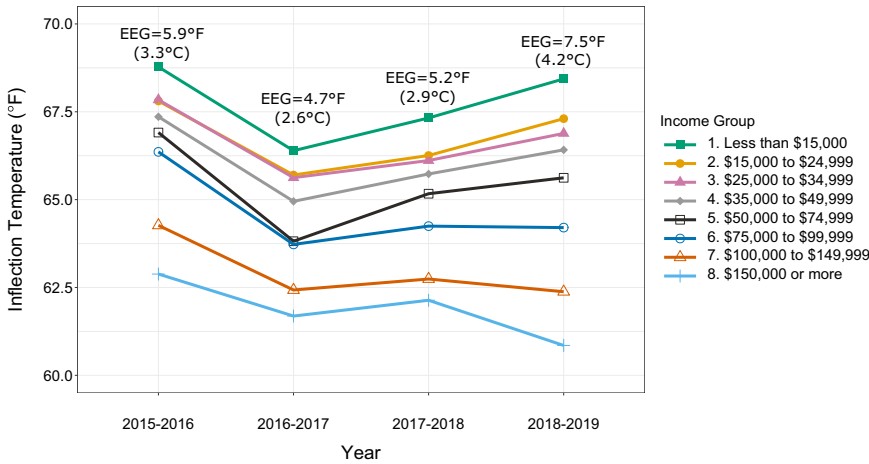

**Fig. 4 Energy equity gap (EEG) and median inflection temperature changes across study years.** Each line represents one income group. Each data point represents the median inflection temperature of the income group for that year.

discomfort becomes too great (high indoor temperature) populations are at a higher risk for heat stroke[49].

The energy equity gap shrinks and widens, resulting from low-income households first lowering then increasing their inflection temperatures while high-income households have a general trend of lowering their inflection temperatures across the years in our analysis. Thus, highlighting increasing energy inequity in the region.

The energy equity gap narrowed by 20.3% between the first two years of our study, but then widened by 10.6% and 44.2% in the last three years of our study, as seen in Table 1. We find that a change in cooling degree days or residential electricity price correlates to energy equity gap changes in the following year.

Between years one and two, there was a 2.4 % increase in residential electricity prices and a 3.6% increase in cooling degree days. This parallels with a 10.6% increase in the energy equity gap in year 3, most likely caused by low-income groups waiting longer to turn on their AC systems. This may signify a delayed price elasticity of demand effects in year-to-year residential electricity price changes and a warming climate. Between years two and three, the residential electricity price rose again by 2.7% and cooling degree days by 2.5%, which corresponds to a 44.2% increase in the year's energy equity gap. Thus, both a higher temperature and a higher electricity price can cause energy equity to deteriorate. Within our study population, low-income

**Table 1 Temperature, electricity price, and energy equity gap shifts in Arizona.**

| Arizona metric | Change from year 1 to 2 | Change from year 2 to 3 | Change from year 3 to 4 |
|---|---|---|---|
| Warmest Month Average Max outdoor temperature | 1.0% | −0.2% | −0.2% |
| Cooling Degree Days (CDD) | 3.6% | 2.5% | −5.2% |
| Average residential electricity retail price (cents/kWh) | 2.4% | 2.7% | −2.7% |
| Energy equity gap | −20.3% | 10.6% | 44.2% |

Warmest Month Average Max is the average maximum temperature of the hottest month in a year.

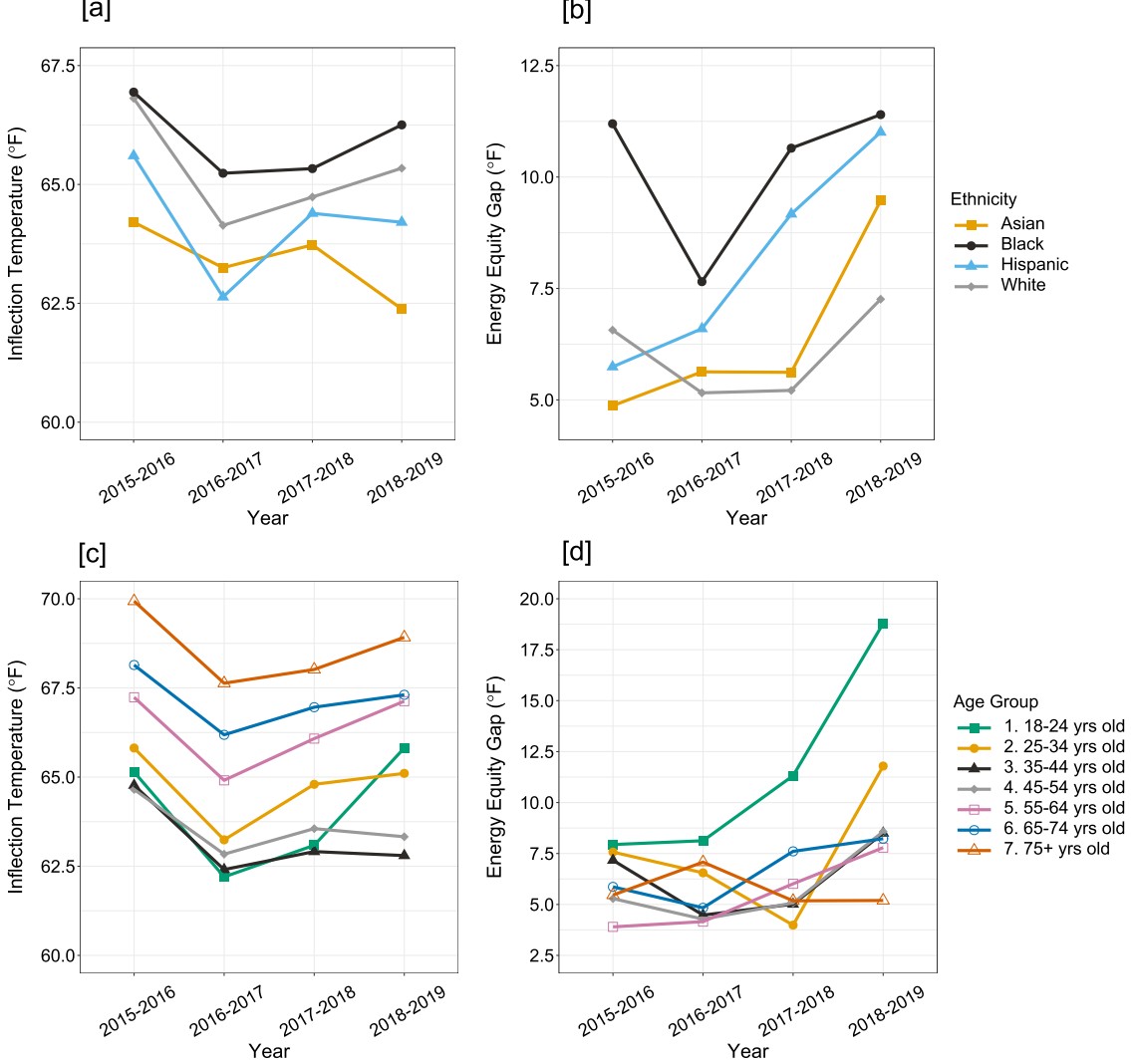

**Fig. 5 Inter- and intra-group comparison of the inflection temperature and energy equity gap for ethnicity and age groups.** Median inflection temperatures by (**a**) ethnicity and (**c**) age group show disparities across demographics. The energy equity gap highlights energy consumption behavior differences between high and low-income populations within their respective (**b**) ethnicity and (**d**) age groups. See Supplementary Information Notes 3 and 4 for more details.

households are more likely to live in older residences (see Supplementary Information Note 2), which can contribute to more significant energy needs and financial strain required to cool the home.

Price shifts will impact electricity consumption shifts within minority groups and those at the intersection of multiple vulnerable groups (e.g., low-income minority groups or low-income elderly populations) differently. Figure 5a, c show the median inflection temperature for each ethnicity and age group; Fig. 5b, d show the energy equity gap within each ethnicity and age group, respectively. Comparing Fig. 5a, b (assuming similar

temperature preference between ethnicities[50,51]), we see that the overall inflection temperature is highest in the Black population. This combined with high energy equity gaps, indicates that the Black population is worse off and experiences high levels of inequity. In the Asian population, the overall median inflection temperatures are low yet there are wide energy equity gaps, indicating high income disparity within the group.

In the Black population, we see increasing disparity followed by the 2.4% electricity price increase from year 1 to 2, resulting in a 39% increase in the energy equity gap from year 2 to 3, earlier than the large price shock that affected the whole population.

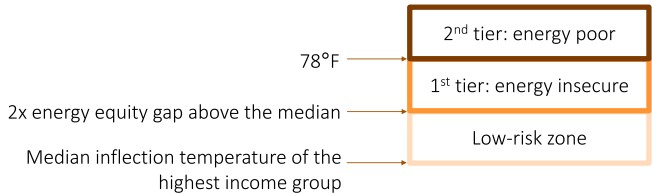

**Fig. 6 Tier systems for energy poverty and insecurity identification using the energy equity gap.** The darker the shade the more severe the level of energy poverty experienced by a household is.

This indicates that the Black population is disproportionately affected by price shifts compared to the other ethnicities. The white population most closely resembles the overall population trend because white populations account for the majority (>70%) of residents in our sample. In general, we find that even when a minority group's median inflection temperature is low, there can be a high disparity between low and high-income populations (see Supplementary Information Note 3), evident in the Asian population.

We also investigated energy poverty and equity across head of household age groups (Fig. 5c, d). There are statistically significant differences between median inflection temperatures across age groups ($p < 0.05$), as confirmed using the Mood's Median test (see Methods), indicating low probability that these variations occur solely due to chance. For the 18–24 age group, both the median inflection temperature (>7 °F, 3.9 °C) and the energy equity gap increased sharply (>14 °F, 7.8 °C) between 2018 and 2019, while for the older populations, the energy equity gap had little change across the years. From an energy poverty targeting standpoint, this highlights that within the elderly population, all residents should be targeted to reduce inflection temperatures, while for the youngest age groups, the most effective poverty eradication policy would be to target low-income groups.

For all age groups except for 75+, the later increase in the energy equity gap is from low-income households getting worse off and high-income households performing better, most evident in age groups 18–24 and 25–34. The difference in inflection temperatures between age groups may be attributed to each age group's different temperature comfort levels. Elders may prefer warmer indoor temperatures, and cooling air from an AC system may inflame arthritis[52], but caution should be used when differentiating between a comfortably warm temperature and one that puts the resident at risk for a heat-related illness[53]. Because there is a significant relationship between household income and inflection temperature, the high energy equity gap in younger age groups may be attributed to larger income inequality among young people.

We acknowledge that there is a chance that the inflection temperature and energy equity gap can be affected by indoor thermostat preferences. To account for varying preferences across ethnic and age groups, we investigate the inflection temperature disparities for different income groups within demographic distinctions (see Supplementary Information Note 3). For example, if one ethnic group preferred to turn on their AC units at a certain temperature, we expect to find a narrow vertical distribution for the inflection temperatures (i.e., the Hispanic population in Supplementary Information Note 3). On the other hand, if the different inflection temperatures represent inequity, we expect to find a wide vertical distribution (i.e., the Black and Asian population in Supplementary Information Note 3). Thus, the energy equity gap can highlight inequities across and within groups in a region. We present a more detailed discussion of the preference limitation in the Supplementary Information Note 3.

We introduce a tiered system (Fig. 6) to identify the households with the highest risk of heat-related illness and death. First, we assume the median inflection temperature of the highest income group is the ideal inflection temperature for this region. This assumption stems from the belief that the highest income groups are the least likely to constrain their budget and thus would initiate cooling systems earliest in the year. Similar to using a standard deviation, we define people with inflection temperatures between one and two energy equity gaps above the ideal inflection temperature to be in the low-risk zone. Next, households with inflection temperatures between two energy equity gaps and 78 °F (25.6 °C) are in the energy insecure zone. Within government buildings, it is recommended that 78 °F degrees be the indoor set point[54], meaning this temperature setting may limit the risk for mold and allergen build-up, as well as heat-related illness and death. Finally, households with inflection temperatures higher than 78 °F (25.6 °C) are defined as energy poor. We use the indoor 78 °F (25.6 °C) comfort set point as our energy poverty threshold because households would need some degree of cooling above this outdoor temperature.

We acknowledge that there are multiple factors that can influence comfort levels and the health risk of occupants in high heat temperatures. Previous studies have shown that heat-mortality risk occurs when outdoor temperature rises above 90 °F (32.2 °C)[55]. However, our goal is to identify households at risk for both health-related illness and death, which can result from a lower temperature threshold. We derive this lower threshold (78 °F, 25.6 °C) from recommended indoor AC setting for government buildings[54,56], as well as from recommendations of various utility companies[57,58]. A key assumption in this threshold is that when the outdoor temperature is above this level, the indoor temperature would rise enough to warrant turning on the AC.

Using this tiered system, policymakers and utility companies can create more targeted weatherization aid programs. When we apply the tiers system to 2015–2016 data (Table 2), we identify 86 energy poor (1.9% of our sample) and 214 energy insecure households (4.7% of our sample).

**Comparing the energy equity gap to income-based measures.** While US government assistance programs lack a clear definition for energy poverty, change in energy burden is often used to measure the outcome of these programs[9,11]. Both LIHEAP and WAP use income limit by household size to determine eligibility[59,60], with some flexibility for states to determine what income level to set as the eligibility threshold. When using the 10% energy burden threshold to identify energy-insecure households in our study region, we found that less than 3% of households were defined as energy insecure (Fig. 7), of which over 70% reside in the lowest income group (<$15,000) for all years in our study. Comparing the energy equity gap categorization with the energy burden measure of individual households, we find that few households (≤ 20) were identified as energy insecure or energy poor under both metrics (Table 2, also see Supplementary Information Note 7 for a visual representation). The energy burden metric categorizes more households as energy insecure, but our tiered system identifies more energy-poor households who may be placing themselves at risk by limiting cooling-associated energy use. The energy burden metric misses more than 95% of those with high inflection temperatures and, therefore, a higher risk of extreme heat exposure. Of that 95%, around half of the households are in one of the three low-income groups (<$35,000). There are energy insecure households identified in the non-low-income groups, which hints that while some

**Table 2 Comparing the energy equity gap (EEG) and financial-based energy assistance categories.**

| (Number of households in each category) | 2015–2016 | 2016–2017 | 2017–2018 | 2018–2019 |
|---|---|---|---|---|
| Total households | 4577 | 4522 | 3852 | 2650 |
| Energy equity gap (EEG) low risk zone | 2719 | 2143 | 1889 | 1619 |
| EEG 1st tier: Energy insecure | 214 | 631 | 484 | 42 |
| EEG 2nd tier: Energy poor | 86 | 83 | 57 | 59 |
| Households with energy burden ≥10% | 141 | 135 | 111 | 88 |
| EEG low-risk zone households with energy burden ≥10% | 94 | 59 | 64 | 55 |
| EEG 1st tier households with energy burden ≥10% | 6 | 16 | 9 | 1 |
| EEG 2nd tier households with energy burden ≥10% | 3 | 4 | 1 | 2 |
| EEG 1st and 2nd tier households with energy burden <10% | 274 | 587 | 286 | 93 |
| EEG 2nd tier households not eligible for LIHEAP | 72 | 63 | 48 | 48 |
| EEG 2nd tier households not eligible for WAP | 53 | 37 | 33 | 29 |

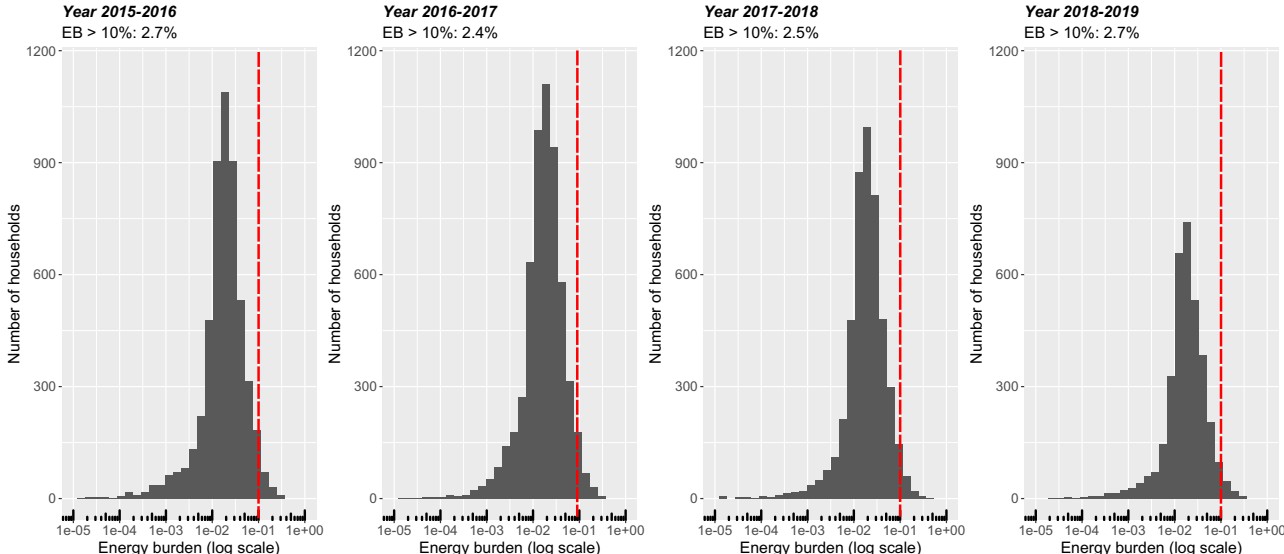

**Fig. 7 Energy insecurity measured using the energy burden (EB) metric.** The x-axis represents the percent of income a household spends on electricity. The red dotted line indicates the 10% income spending threshold, and EB10 details the proportion of households above the energy spending threshold. For example, in 2015–2016, 2.7% of households in our sample spent more than 10% of their income on electricity. The maximum amount of spending in our sample population is 35%.

households are not classified as low-income, they may have low disposable income (e.g., high mortgage or rent costs) and limit their financial burden by reducing their energy consumption. This further highlights the need for multiple energy poverty and insecurity measures to determine financial and behavioral energy consumption trends in energy-insecure households.

We also compared eligibility for LIHEAP and WAP and households in the 2nd tier of the energy equity gap metric. In our dataset of 6002 households, 871 qualified for LIHEAP and 1553 qualified for WAP using their respective income threshold[9]. We found that 48–72 households each year have high inflection temperatures but are not eligible for LIHEAP. Because WAP has a higher income threshold than LIHEAP, all households that qualify for LIHEAP also qualify for WAP, so this range reduces to 29–53 households when comparing the energy equity gap and WAP. Many of these missed households are just on the edge of the low-income threshold but have uncomfortably and sometimes dangerously high inflection temperatures, making them vulnerable without assistance.

**Limitations of analysis and opportunities for future work.** Energy poverty exists in multiple forms, leading to numerous

limitations in any quantification method. Here we present a discussion of the limitations of our methods and opportunities for future improvements. Uncertainties in using the iteration of the energy equity gap outlined in this paper include the lack of heating data from the natural gas provider. From the dataset provided by the electric utility, we gather that 60% of households in this study use electricity for both cooling and heating, while the remaining most likely use natural gas or oil for heating. However, we did not find the type of heating system to be a significant indicator of household inflection temperatures. Thus, the model used to calculate the inflection temperature still stands for this particular electricity-based dataset (see Methods), particularly for a high-heat area like Arizona where heat-related illness and death is significantly higher than cold-related ones[40,41].

Housing characteristics that relate to the energy efficiency of the home[61] (e.g., number of windows, insulation, wall thickness, finishing material, the orientation of the home, etc.) were not included in the dataset but would be valuable additions to future utilities data collection effort. We did find a relationship between residence age and income group, where a large proportion of low-income households lived in older homes (see Supplementary Information Note 2), which could contribute to higher inflection temperatures and less overall household energy efficiency.

One limitation of this study is a lack of indoor thermostat data and an inability to quantify the actual indoor temperature of homes. Thermostat data could provide more information about the willingness of occupants to consume energy for cooling and heating needs. However, thermostat information would not reveal a consumer's true temperature comfort preferences. The inability of thermostat data to identify true household preferences results from household occupants adjusting their thermostat based on multiple factors such as spending limits (i.e., disposable income), comfort, and energy conservation habits. Thus, a person might deliberately keep their thermostats at non-ideal temperatures (i.e., energy limiting behavior), rather than their true comfortable temperature, to save money or energy. Another possibility is that the energy equity gap could be wider than our analyses suggest due to the urban heat islanding effect[62,63], with low-income households being more exposed to high heat due to less shading and vegetation in their urban environments.

In spite of these limitations, we believe this analysis provides a crucial first step in identifying energy limiting behavior in low-income households in a metropolitan region. A fruitful direction for future work would be to investigate actual temperature discrepancies within the home.

## Discussion

The energy burden metric targets households who limit other necessities to meet energy needs but misses out on those who limit energy spending to meet other needs. The energy equity gap fills this void by identifying who chooses to endure higher temperatures in the summer and how their behavior may change due to price spikes and weather changes. The two types of metrics should be used in conjunction to identify households experiencing multiple forms of energy poverty: those who experience financial strain while satisfying their energy needs, and those who forgo energy consumption to reduce financial strain and satisfy other necessities. By considering behavior patterns in addition to spending patterns, policy makers will be able to better identify who is in need of energy assistance. As nations continue designing equitable and sustainable energy policies, regions need a straightforward method to evaluate the current level of energy poverty from an economic, consumption, and behavioral perspective.

By grouping households by income and energy equity gap severity, policymakers and utility companies can target the most at-risk households requiring urgent financial help with energy bills and weatherization. The value of the energy equity gap is that it creates the possibility of a sliding-scale energy poverty assistance program, which could set weatherization targets for the population based on their ability to adapt to extreme weather events (e.g., heatwaves) and how they are performing relative to others in the region. For example, if primary policies were designed to target households with an inflection temperature above 78 °F (25.6 °C) and those that spend more than 10% of their income on meeting their energy needs, this could reduce financial strain and risk of heat-related illnesses in the region. Secondary policy targets should focus on households with low income and above-average inflection temperatures. These households are likely to suffer from multiple forms of energy poverty and insecurity but are not at high risk for heat-related death.

On the other hand, high-income households with high inflection temperatures may be best suited for discounted weatherization programs. Despite having higher incomes, weatherization costs may still be too high for these households if they have limited disposal income. When adapting the energy equity gap to other regions, electricity consumption may be sufficient for similarly high-heat regions. However, researchers and energy planners should consider gas, electric, and potentially other fuels used for heating to estimate total energy consumption for colder climates. Calculating the energy equity gap requires the same information for the traditional energy poverty metric, so the cost to compute and utilize the energy equity gap would be marginal.

In the ever-evolving discussion around equity, justice, and policy, we need to study and develop policy that answers the needs of those historically and systemically marginalized. We can start by identifying those falling through the policy cracks of economic-based poverty metrics by casting a finer net which also includes poverty displayed through energy consumption behavior. The energy equity gap contributes to the discussion of existing energy poverty metrics by capturing a region's relative progress while including the households that income-based metrics may have left behind. By targeting the population with higher-than-ideal inflection temperatures with equity-centered policies, regions will more effectively eradicate energy poverty and assist their residents in adapting to climate change.

## Methods

**Data**. The data was provided by Salt River Project, a large utility company in Arizona. The dataset comprises two parts: first, hourly electricity consumption in kWh from May 2015 to April 2019 for 6000 households and the billing plan for each household; second, a comprehensive Residential Equipment and Technology Survey conducted in 2017 for those households. The survey included information on household sociodemographic information, and dwelling characteristics (e.g., residence age, size, and type). For summaries of demographics information please see Supplementary Information Note 1.

Salt River Project provides different billing packages from which customers can choose based on their preferences. Each billing package has its own pricing rules and condition. The billing packages can be categorized into Basic Rate Plan, Time-of-Use Plans, and Prepaid Plans. Pricing rules were integrated into the consumption dataset to account for energy consumption patterns based on electricity pricing, where a uniform weighted average electricity price is calculated across 24 h of the day.

The hourly electricity consumption was aggregated into daily consumption. Daily consumption information was then coupled with daily average temperatures for the study region, compiled from WeatherForYou.com[64].

**Inflection temperature**. The inflection temperature is defined as the outdoor temperature where a household shifts from using its heating system to its cooling system. We recognize that there may be a temperature range where the household uses neither heating nor cooling, and the base level energy consumption during that period would be temperature-independent[46]. In this context, the inflection temperature is still an indicator of the shift in energy consumption behavior. A household's inflection temperature is calculated using a nonlinear regression model (Eq. (1)), which estimates daily electricity consumption of household $i$ on day $t$ ($E_{i,t}$) based on the following variables: daily average temperature ($T_t$), electricity price based on the billing plan of the household and season ($P_{i,s}$), dummy variables of whether day $t$ is a holiday ($H_t$), day-of-the-week fixed effects ($\delta_t$), and month-of-the-year fixed effects ($\mu_t$). When modeling day-of-the-week and month-of-the-year dummy variables, Wednesday and March were dropped, respectively, to prevent collinearity.

$$E_{i,t} = \alpha + \beta_1 \times T_t + \beta_2 \times T_t{}^2 + \beta_3 \times P_{i,s} + H_t + \delta_t + \mu_t \quad (1)$$

The quadratic equation models the relationship between daily electricity consumption and daily average temperature. We chose a quadratic relationship because it best coincides with the shape of the electricity consumption and temperature data, and a median $R^2$ value of 0.8 for all households (Fig. 2). The convex shape of the curve confirms the notion that electricity consumption is highly correlated with temperature. The inflection temperature is the minimum electricity consumption point (Eq. (2)) and signifies the outdoor temperature a household must experience before initiating their AC units.

$$T_{\inf} = T_t \; when \; f\prime(E_{i,t}) = 0 \quad (2)$$

We acknowledge that in building literature there are studies that use a piecewise linear function to identify cooling and heating turn on points (often referred to as balance points)[45,46,65–67]. We use the quadratic function over the piecewise linear function due to higher $R^2$ values, which is consistent with other studies[68] (see Supplementary Information Note 12 for more discussion).

The outliers of the inflection temperature model are defined as any household with an inflection temperature below 30 °F (−1.1 °C) or above 120 °F (48.9 °C), based on outdoor temperature limit ranges measured within the study region. An inflection temperature outside of this bound may indicate incomplete electricity consumption data. Within our study, we filtered out 0.5% in year one, 1.6% in year two, 1.2% in year three, and 0.2% in year four from our analysis due to their classification as outliers. There are a total of 6002 households across the four years of study, but not all households have complete data for all years, which is why the total number of households in Table 2 decreased. One reason for the incomplete data may be that households started or stopped service midway with this utility company.

**Computing the energy equity gap.** The energy equity gap quantifies the relative energy consumption behavior differences between low and high-income groups using the inflection temperatures. We hypothesized that lower-income households are more likely to have higher inflection temperatures due to financial limitations and a desire to delay cooling their homes to reduce their energy burden (i.e., percent of income spent on energy services). After calculating the inflection temperature ($T_{inf}$) for each household for one year, we group households by income. The energy equity gap for year y, $G_y$, is the maximum median inflection temperature ($\max(T_{inf,median})$) minus the minimum inflection temperature ($\min(T_{inf,median})$) among all income groups.

$$G_y = \max(T_{inf,median}) - \min(T_{inf,median}) \quad (3)$$

We hypothesized that lower-income households are more likely to have higher inflection temperatures. To test our hypothesis, we performed tow-tailed Mood's Median tests, a nonparametric alternative to a one-way analysis of variance, for significance. A significant result from a Mood's Median test demonstrates that one sample stochastically dominates another, and the differences between sample medians are statistically different. Tests were also performed on median inflection temperatures of ethnicity and age groups, with P-values shown in Table 3. When we group the households by income, we see significant P-value results for all four years, which indicates that the difference in median inflection temperatures of income groups have a close to 0% chance of solely being random (i.e., they are statistically significant). We see similar results when we group the sample population by age, which means age may also be a strong indicator of inflection temperature. Therefore, we cannot rule out that age may play a role in electricity consumption habits (e.g., older people may prefer to turn on their AC at a higher temperature), which would affect the inflection temperatures seen across groups. That being said, when computed within an age or ethnicity group, the energy equity gap can highlight when members are experiencing worsening poverty (i.e., the gap is widening), or when members of the group are adapting to temperature changes in a similar fashion (i.e., the gap is narrowing). While the ethnicity p-values are not on the same order of magnitude as income or age groups, we find that there is less than a 1% chance that the variation between ethnic groups is solely due to chance for years one, two, and four, and less than 6% for year three, thus indicating high statistical significance.

We also considered the potential effects of type of residence (i.e., single-family home, multi-family home, condo, mobile home, townhouse), residence age, and residence size that can have on the household's inflection temperature. However, we find including these variables would introduce multicollinearity into the model because they are correlated with income. For more details see Supplementary Information Note 8.

**Computing the traditional energy poverty metric.** The traditional economic based energy poverty metric, energy burden, is defined as the percent of income a household spends on satisfying their energy (e.g., electricity) demand. We calculate the proportion of energy expenditure of each household for each year using income, residential electricity price, and energy consumption. For each household, the utility company provides the income bracket each household falls into. We use the midpoint of each income group to estimate the percent of income spent on energy consumption. For the lowest income group (<$15,000), $10,000 was taken as the midpoint; for the highest income group (>$150,000), $175,000 was taken as the midpoint.

$$S_{i,y} = \frac{\sum E_{i,t} \times P_i}{I_{i,m}} \quad (4)$$

$S_{i,y}$ is the energy expenditure over income of household $i$ in year $y$, $E_{i,t}$ is the daily electricity consumption of household $i$ on day $t$. $P_i$ is the average electricity price of the billing plan of household $i$. $I_{i,m}$ is the midpoint estimate of income for household $i$.

**Reporting summary.** Further information on research design is available in the Nature Research Reporting Summary linked to this article.

## Data availability
The raw and processed electricity consumption data and the residential energy technology survey data are available under restricted access bound by a non-disclosure agreement, access can be obtained by upon reasonable request to the authors and with permission from the Salt River Project. Records of mean daily outdoor temperatures were retrieved from WeatherForYou.com by way of web scraping and can be accessed here. Source data for Fig. 3, Supplementary Information Figs. S3, S6, and S9 can be accessed here.

## Code availability
All data and models are processed in Python 3.8.5. The figures are produced in PowerPoint and R studio (based on R 4.0.3). All custom code[69] is available on GitHub at https://github.com/Pa223/The-Energy-Equity-Gap.

### Table 3 P-values from two-tailed Mood's Median tests on median inflection temperatures of income, ethnicity, and age groups.

| Grouping | 2015-2016 | 2016-2017 | 2017-2018 | 2018-2019 |
|---|---|---|---|---|
| Income | 5.36E−23*** | 6.85E−22*** | 2.50E−16*** | 2.05E−13*** |
| Ethnicity | 5.94E−03* | 6.02E−05*** | 2.85E−01 | 1.73E−02. |
| Age | 2.30E−29*** | 1.61E−32*** | 8.91E−20*** | 7.36E−17*** |

Alpha = 0.05.
Signif. codes: "***" [0, 0.001] "*" (0.01, 0.05] "." (0.05, 0.1] " " (0.1, 1].

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

## Acknowledgements

This work is funded by the National Science Foundation [grant number 2029511 (Y.Q., D.N.); 2017789 (D.N.); 1757329 (Y.Q.)]. We thank graduate research assistant Ali Iftikhar for his support in initial regression analysis, and Jiehong Lou for her support of using the high-performance computing cluster. We thank Alex Davis, Baruch Fischhoff, Granger Morgan, and our other colleagues in the Department of Engineering and Public Policy at Carnegie Mellon University for providing valuable insight and feedback. Nock also acknowledges support from the Google Award for Inclusion Research and the Scott Institute for Energy Innovation, where she is an energy fellow.

## Author contributions

D.N. conceived the research idea and designed and oversaw the research process. S.C. designed and performed the analysis. S.C. wrote and revised the initial draft of the paper. D.N. and Y.Q. reviewed and revised the paper. B.X. collected and cleaned the data. Correspondence and requests for materials should be addressed to D.N. or Y.Q.

## Competing interests
The authors declare no competing interests.

## Additional information

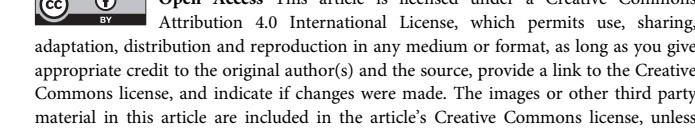

