## [Peer Review File · Nature Communications]

Reviewer comments, first round

Reviewer #1 (Remarks to the Author):

Title: The Energy Equity Gap: Unveiling Hidden Energy Poverty

This article introduces a new method for estimating energy-limited behavior in low-income households using a residential electricity consumption dataset. Although the authors present a new metric for measuring energy poverty, there are concerning measurement and conceptualization issues. I do appreciate the authors' efforts in proposing a new measure.

1. Conceptualization issue: Although this is a new concept that the authors claimed income-based energy poverty metric misses people's behavior, I would argue that income-based energy poverty implies the level of low-income households' energy consumption reflecting from the utility bills and low-income households' energy behavior. More importantly, energy poverty needs to measure utility bills; otherwise, it won't be called energy poverty. Thus, this article's measure only reflects energy behaviors, not is classified as energy poverty. Additionally, energy insecurity measures should reflect the impacts of energy accessibility or energy hardship issues.

2. Introduction (ln 33): The statement of "A large portion of these deaths could have been prevented if people could cool their homes properly" might need to modify because many low-income households do not own an AC.

3. After reading the long introduction on the existing energy poverty metric, the authors' definition of behavioral-based energy poverty is unclear and well-defined.

4. There are some measurement issues relating to the metric:

(1) Climate or outdoor temperature can only make energy poverty worse or better by considering house insulation and energy appliances in addition to energy usage (temperature setting). Overall, only considering the temperature misses the opportunity to consider relevant factors (electricity prices, household characteristics, housing conditioning, indoor temperature setting, etc.). All of these factors should be considered in one model.

(2) Income groups should be considered the household size.

(3) There are some sampling issues. It seems like each year has different households, so it is difficult to compare across the years. Fig. 4 should also show the summer and winter differences.

(4) LN 251 on page 10 and Table 1: I would like to know the reasons for having significant changes in retail electricity prices and cooling degree days in Years 3 and 4. Also, is Max temperature is from indoor or outdoor?

(5) One of the most significant issues for this article's measure is the lack of an indoor temperature setting.

5. Energy insecurity comparison – it would be interesting to compare those households that forgo energy to meet other necessities vs. their counterparts.

Reviewer #2 (Remarks to the Author):

This paper presents a novel approach to quantifying energy poverty by defining an energy equity gap and comparing the gap across income groups. The paper is well written with a thorough review of the literature, clear evidence and nicely presented methods. The authors make a strong case for the policy relevance of their approach by demonstrating how their metric identifies households that may forgo energy for other needs, which are not captured by the energy poverty metrics currently in use. That said, there are a few points in the paper that require clarification. So, I recommend the paper for publication with minor revisions. I list the points that need to be addressed below.

Line 65: county  country

Fig 1: A picky point about the placement of surveys on the 2x2 plot. As I understand it, survey-based data isn't necessarily relative. It can be used to determine whether a family is above or below an absolute threshold.

Line 93: The authors should consider adding a footnote explaining the difference between equity and equality - some readers may not be familiar with the distinction.

Line 94: is  are

Line 103: delete the 't'

Line 190-191: At this point I started to wonder whether the authors had any way to control for housing type (e.g. Multi-family vs. single), family size, or personal preference? The authors mention "thermostat preferences" on p. 13 and mention housing characteristics in their discussion of "Limitations", but perhaps it would be worth raising these issues earlier in the paper so that readers are aware that the authors have considered them.

Line 236: The authors wrote "As the energy equity gap shrinks and widens, we see the driving factor is that low-income households are getting worse off while high-income households are improving". Causation implied by this sentence isn't totally clear. Are the authors saying that the widening EEG leaves low-income people worse off? It might be more accurate to say that a widening EEG exposes people to greater risk.

Table 1: The largest change in EEG occurred between Year 3 and 4, but price and CDDs had both declined. Perhaps worth acknowledging this anomaly.

Line 277: This page has the first mention of age. It appears to be an important covariate. However, the authors do not explain what age they use in the analysis. I assume it is the age of the household head, but it should be explained clearly.

Line 278: E-23 should be a number like 0.05?

Lines 308-317: I appreciate that the authors attempted to address individual preferences. However, it isn't clear to me why preferences would cluster by ethnicity. Has this been shown in prior research? Should we also expect to see differences within ethnic groups?

In addition, in this section, the authors describe several patterns they observe between different ethnic groups. Specifically, they note how Black and Asian HHs have a wide vertical distribution of inflection temperatures, while Latino HHs have a narrower distribution. When I visually inspect the Figs S4-6, I see that the widest distribution of Inflection Temp among Latino HHs (~10 deg F in 2018/19) is wider than the distribution among both Black and Asian HHs in some years. Are there statistical tests that the authors can use to show that the variation between groups they describe is significant? The SI skirts the issues. The authors state (SI Lines 78-79) "Black and Asian groups may be less tolerant of 78 heat, therefore have lower inflection temperatures overall" but they don't specify any statistical test. This leads me to believe that it isn't a significant relationship. If this is the case, the authors should note that the pattern exists, but is not significant and their discussion is speculative.

Lines 330-331: Here the authors discuss their selection of a "comfort set point" of 78F, above which HHs are defined as energy poor. For this they cite a study by Díaz et al. which found mortality risk increases above a daily max temp of 90F (actually they find risk really jumps at a max of 36.5F or ~98F). The authors should note that the study they cite refers to daily max temp, which is quite different than their comfort set point, which would theoretically be maintained for the entire day. Of course, their point of health-related illness vs. mortality still stands.

Figure 7: Is the number at the top of each graph the mean, median, or some other metric. I did

not see it specified anywhere.

Line 400: This is the only mention of SDGs in the entire document. Though they apply globally, the concept is generally used in the context of developing countries. I am not arguing that it doesn't apply to their analysis; however, none of the indicators for SDG7 specifically address the type of energy poverty that the authors present in this paper. If the authors do want to refer to the SDGs, I suggest they define SDG7 and add a short discussion in the intro explaining to readers why it's relevant for this context even if this type of energy poverty isn't specifically addressed by the existing targets and indicators.

Reviewer #3 (Remarks to the Author):

An interesting paper with some good analysis. There does seem to be some limitations which are mostly described in the Limitations section. Some detailed questions/comments:

Line 65: "country-to-county" is this intended to have a second "r" to make it "country"

Line 76: "This metric does not assume..." Could you clarify which metric you are referring to – it seems to contrast to the prior sentence.

Line 103: "t households ": typo?

Line 107-108: "energy poverty analysis in developed countries has been dominated by absolute-primary economic-based metrics" perhaps this is somewhat overstated as there are many studies on subjective indicators (some that you cited and many others)

Line 114: "First, they are sensitive to energy prices and" is this a repeat of line 100?

Line 123-124: "unsafe practices (e.g., using stove or space heaters to heat space" space heaters to heat space sounds like an intended activity which is not necessarily unsafe.

Line 126-129: "we find a void of metrics that can identify energy-limiting households (i.e., those without comfortable indoor temperatures) who may put themselves at risk of heat-related illness, excess indoor moisture, mold growth, and other adverse health effects" Similar to the point for line 107-108, I think there are many papers on this.

Line 148-149: "effect of weather on energy expenditure by examining trends in socioeconomic and ethnic groups." The phrases may not fit together all that well.

Line 150-151: "eliminates the effect of weather or temperature on electricity usage" outdoor temperature rather than indoor

Line 155: "identify energy insecure households that are dangerously close to sinking into energy poverty" perhaps a reminder of how you are defining energy poverty could be useful

Line 162: "assuming there is no difference in comfort preference or need across income groups" are these realistic assumptions and could there be large differences across groups or individuals? Do you control for age of household occupants in the main analysis? Perhaps older occupants have different income and also different heating preferences. Also for building characteristics, e.g. insulation.

Line 195: "until later in the summer" how do you know it's later in the summer – are the temperatures consistently hotter in late summer without much variation between days?

Line 235-236: "As the energy equity gap shrinks and widens, we see the driving factor is that low-income households are getting worse off while high-income households are improving." Perhaps there is scope to improve this sentence. Is it focusing on changes over time?

Line 244-245: "This may be caused by the delayed price elasticity of demand effects in year-to-year residential electricity price changes and a warming climate" Can it also just be random variation from the comparison of point estimates?

Line 260-317: is the sample size large enough for this analysis? There are some large changes between years in Figure 5. Are all the variables e.g. age, income, ethnicity included as controls in a single estimation at any point of the paper?

Line 361: "dangerously high inflection temperatures" does this depend on characteristics of the occupants? E.g. some high temperatures may not be a big deal for young and healthy occupants

Line 422: "identifying those falling through the policy cracks by casting a finer net." Is it finer or just different?

Line 500: Does Table 3 mean that all groups are different to each other e.g. all 8 income groups are different to all of the others? Are there more details on the stars? Are the income, age, and ethnicity variables used in each part of the analysis, or are they included just separately?

Energy Equity Gap: Unveiling hidden energy poverty

Response to reviewer comments*

***Reviewer comments are indicated in black, our responses in blue, and our text additions in red.**

References are at the bottom of each page

Reviewer 1

This article introduces a new method for estimating energy-limited behavior in low-income households using a residential electricity consumption dataset. Although the authors present a new metric for measuring energy poverty, there are concerning measurement and conceptualization issues. I do appreciate the authors' efforts in proposing a new measure.

Thank you for your thorough review and thoughtful suggestions, below are our responses and edits made according to your comments. Thank you for helping us improve our paper!

1) Conceptualization issue: Although this is a new concept that the authors claimed income-based energy poverty metric misses people's behavior, I would argue that income-based energy poverty implies the level of low-income households' energy consumption reflecting from the utility bills and low-income households' energy behavior. More importantly, energy poverty needs to measure utility bills; otherwise, it won't be called energy poverty. Thus, this article's measure only reflects energy behaviors, not is classified as energy poverty. Additionally, energy insecurity measures should reflect the impacts of energy accessibility or energy hardship issues.

a) First, we would like to clarify some definitions of energy poverty:

- i) As you mentioned one common way to measure energy poverty is through energy bills, often called energy burden. Energy burden is a bivariate measurement which shows the percentage of household income that is spent on energy bills¹. By itself, it is not an energy poverty measure but an indication of expenditure on energy bills. Once a threshold is set (i.e., 10% or 6%) then falling above or below that line can indicate who may be experiencing energy poverty. As the reviewer mentioned, it is also true that the income-based energy poverty metric does imply low-income households' energy behavior, but it only reflects those who limit other necessities to meet energy needs (and thus higher percentage of

¹ Progress in Energy High energy burden and low-income energy affordability: conclusions from a literature review. (2020) doi:10.1088/2516-1083/abb954.

- income spent on energy) but misses out on those who limit energy spending to meet other needs.
- ii) Another more common definition of energy poverty being used in the existing literature is a lack of access to modern energy services. It can also refer to a situation where a household does not have access or cannot afford to have the basic energy or energy services to achieve day to day living requirements². To your point percent of income spent on bills can address this once a baseline for “affordability” is established, but there is also a need to quantify who cannot afford to purchase electricity.
 - iii) Energy affordability is the ability to pay for the quantity of energy required for basic needs, not infringing on the affordability of other necessities. Thus, households who forgo energy to reduce financial strain (e.g., not fixing a broken air conditioner or heating system) may still be experiencing an energy deficit, even though it may not be reflected in their total energy expenditure.
 - iv) We have clarified these definitions in the main text as follows (See line 147 in the main text):
“We define energy poverty as having inadequate energy services within the household, or an inability to consume energy at a desired level³. Thus a more holistic definition of energy poverty would include people who limit their energy consumption (i.e., display energy limiting behavior), and those who spend a large portion of their income on their energy bills (i.e., high energy burden).“
- b) While energy poverty has traditionally been defined in economic terms in developed countries⁴⁵, there is evidence that poverty includes more facets of a person’s life than solely their energy expenditures⁶⁷. If energy poverty only focused on energy expenditures then those in developing countries who do not have any supply of energy would not be considered energy poor, because

² Doukas, H. & Marinakis, V. Energy poverty alleviation: effective policies, best practices and innovative schemes. <https://doi.org/10.1080/15567249.2020.1756689> **15**, 45–48 (2020).

³ Doukas, H. & Marinakis, V. Energy poverty alleviation: effective policies, best practices and innovative schemes. <https://doi.org/10.1080/15567249.2020.1756689> **15**, 45–48 (2020).

⁴ Romero, J. C., Linares, P. & López, X. The policy implications of energy poverty indicators. *Energy Policy* **115**, 98–108 (2018).

⁵ Sovacool, B. K. Fuel poverty, affordability, and energy justice in England: Policy insights from the Warm Front Program. *Energy* **93**, 361–371 (2015).

⁶ Spiliotis, E., Arsenopoulos, A., Kanellou, E., Psarras, J. & Kontogiorgos, P. A multi-sourced data based framework for assisting utilities identify energy poor households: a case-study in Greece. <https://doi.org/10.1080/15567249.2020.1739783> **15**, 49–71 (2020).

⁷ Piachaud, D. Problems in the Definition and Measurement of Poverty*. *Journal of Social Policy* **16**, 147–164 (1987).

the numerator of energy burden, energy expenditure, would be close to zero, either due to lack of energy services available for purchase to begin with, or a lack of room in the budget for energy services. In a broad sense, energy poverty can also include an inability to consume an adequate amount of energy⁸. Additionally, Piachaud defines poverty as including behavior within a household, such as food preparation and consumption, and budgeting behavior¹. Excluding behavior from definitions of poverty misses the fact that what a person spends on their energy bills depends on their disposable income, indoor comfort, and willingness to spend on their day to day activities^{9,10}. Thus the current definition of energy poverty based on income and utility bills is not inclusive^{11,12}, and including the behavior of households in definitions of energy poverty will create a more holistic approach to poverty alleviations.

- i) We have highlighted the complementary nature of the energy equity gap as follows (See line 411 of the main text): “The energy burden metric targets households who limit other necessities to meet energy needs but misses out on those who limit energy spending to meet other needs. The energy equity gap fills this void by identifying who chooses to endure higher temperatures in the summer and how their behavior changes due to price spikes and weather changes. The two types of metrics should be used in conjunction to identify households experiencing multiple types of energy poverty, those who forgo other needs to meet their energy demand, and those who forgo energy to meet other necessities. By considering behavior patterns in addition to spending patterns, we are able to keep at-risk populations from falling through the cracks of policies intended to help them.”
- ii) Additionally we have enhanced our discussion about energy burden and added the following (See line 112 in the main text): “The energy burden alone is not an energy poverty indicator. However, the relative level of energy burden experienced by a household compared to other residents in the region can be used as an energy poverty metric. Energy burden can be dependent on a number of factors such as electricity price, prices of other goods, and

⁸ Piachaud, D. Problems in the Definition and Measurement of Poverty*. *Journal of Social Policy* **16**, 147–164 (1987).

⁹ Pachauri, S. & Spreng, D. Measuring and monitoring energy poverty. *Energy Policy* **39**, 7497–7504 (2011).

¹⁰ Herrero, S. T. Energy poverty indicators: A critical review of methods. *Indoor and Built Environment* **26**, 1018–1031 (2017).

¹¹ Pachauri, S. & Spreng, D. Measuring and monitoring energy poverty. *Energy Policy* **39**, 7497–7504 (2011).

¹² Herrero, S. T. Energy poverty indicators: A critical review of methods. *Indoor and Built Environment* **26**, 1018–1031 (2017).

heating or cooling needs¹³. The advantage of this threshold metric is that it indicates the economic burden of meeting energy needs and does not have a high computational burden. However, limitations of using the energy burden threshold metric include not distinguishing between gross (i.e., pre-tax) and disposable income (i.e., post-tax and other mandatory charges like mortgage and rent) and not considering local and current costs of living.”

- c) Secondly, hidden energy limiting behaviors are partially driven by disposable income and utility bills. Thus our metric indirectly and partially reflects constraints from utility bills. We do not argue that the energy equity gap should be the sole metric. Instead, we suggest that the energy equity gap is complementary to income-based metrics. When combining the energy equity gap metric and income-based metrics, policy makers could capture people experiencing both types of energy poverty: 1) those who are forced to spend a large portion of their income on energy due to low income, and 2) those who consciously lower their energy consumption to save money, putting themselves below the energy poverty line in income-based metrics. We argue households in the latter category should also be considered energy poor, because they could benefit from being able to consume more energy, or having more energy efficient insulation and appliances to make their indoor environment more comfortable and healthier¹⁴¹⁵.
- 2) Introduction (In 33): The statement of “A large portion of these deaths could have been prevented if people could cool their homes properly” might need to modify because many low-income households do not own an AC.
 - a) We acknowledge that those who do not own an AC are not able to cool their homes properly in a place like Arizona. We have modified the sentence as follows (See line 35 in the main text): “A large portion of these deaths could have been prevented if people could cool their homes properly. We acknowledge that proper cooling ability includes being able to acquire and sufficiently use an AC system.”
- 3) After reading the long introduction on the existing energy poverty metric, the authors’ definition of behavioral-based energy poverty is unclear and well-defined.
 - a) We are grateful for the help in clarifying our definitions. We have clarified the definition of energy poverty (See line 147 in the main text): “We define energy poverty as having inadequate

¹³ Progress in Energy High energy burden and low-income energy affordability: conclusions from a literature review. (2020) doi:10.1088/2516-1083/abb954.

¹⁴ Reames, T. Targeting energy justice: exploring spatial, racial/ethnic and socioeconomic disparities in urban residential heating energy efficiency. *Energy Policy* **97**, 549–558 (2016).

¹⁵ Bednar, D. J. & Reames, T. G. Recognition of and response to energy poverty in the United States. *Nature Energy* vol. 5 432–439 (2020).

energy services within the household, or an inability to consume energy at a desired level¹⁶. Thus a more holistic definition of energy poverty would include people who limit their energy consumption (i.e., display energy limiting behavior), and those who spend a large portion of their income on their energy bills (i.e., high energy burden).”

- 4) There are some measurement issues relating to the metric:
- a) Climate or outdoor temperature can only make energy poverty worse or better by considering house insulation and energy appliances in addition to energy usage (temperature setting). Overall, only considering the temperature misses the opportunity to consider relevant factors (electricity prices, household characteristics, housing conditioning, indoor temperature setting, etc.). All of these factors should be considered in one model.
 - i) Our regression model includes daily electricity prices based on the pricing plan each household uses. The prices may vary day to day for a household depending on the season, weekday or weekend, and whether it’s a holiday. We acknowledge that factors like housing characteristics (insulation, drafty windows, etc.) and household characteristics (income, number of people in a household, etc.) will affect how people use energy. However, our inflection temperature regression is run on individual households. Since these household factors are uniform across the entire time period for each household in our dataset, including them in the temperature response function model does not yield any effects, and these variables have coefficients of zero (See Table A below). Or in other words, these factors will be omitted in the regression model at the individual household level since there is no variation of these variables. We have added Table A in the Supplementary Information Table S6.
 - ii) When we included housing (heating type, number of ACs, AC type) and household characteristics (size, age, and members) we did not see an improvement in the model fit, evident in the zero coefficients of the added factors (see Table A below). Thus we exclude these factors from the inflection temperature regression model because household factors that are uniform across the modeling horizon will not affect model outputs. In the main text we highlight this point (Equation 1) as follows (see line 468 in the main text): “A household's inflection temperature is calculated using a nonlinear regression model (equation 1), which estimates daily electricity consumption of household i on day t ($E_{i,t}$) based on the

¹⁶ Doukas, H. & Marinakis, V. Energy poverty alleviation: effective policies, best practices and innovative schemes. <https://doi.org/10.1080/15567249.2020.1756689> **15**, 45–48 (2020).

following variables: daily average temperature (T_t), electricity price based on the billing plan of the household and season ($P_{i,s}$), dummy variables of whether day t is a holiday (H_t), day-of-the-week fixed effects (δ_t), and month-of-the-year fixed effects (μ_t). When modeling day-of-the-week and month-of-the-year dummy variables, Wednesday and March were dropped, respectively, to prevent collinearity.

$$E_{i,t} = \alpha + \beta_1 \times T_t + \beta_2 \times T_t^2 + \beta_3 \times P_{i,s} + H_t + \delta_t + \mu_t \quad (1)''$$

Table A. Coefficients of regression model for calculating inflection temperatures. The following factors were added to Equation 1 in the main text and did not change nor improve model fit: home heating type (VHOMEHEAT), home AC type (VACTYPE), number of AC units (VACUNITS), residence size (VSQFEET), residence age (VRESAGE), age of the head of household (VHHAGE), the number of people in the household (VHOUSEHOLD_INT), and household income (VINCOME). All but residence size (VSQFEET), residence age (VRESAGE), age of the head of household (VHHAGE), the number of people in the household (VHOUSEHOLD_INT), and household income (VINCOME) are dummy variables. R squared and the coefficients of the original factors included in the model have not changed (coefficients of holidays, weekly and monthly fixed effects are not shown)

Household	r_squared	Coefficients														
		B_temp_avg	B_temp_avg_sq	B_elec_cost	B_VHOMEHEAT_ELEC	B_VACTYPE_GAS_HEAT	B_VACUNITS_TWO	B_VACUNITS_THREE_PLUS	B_VSQFEET_1000_1499	B_VSQFEET_1500_1999	B_VSQFEET_2000_2999	B_VSQFEET_3000_3999	B_VHOUSEHOLD_INT	B_VRESAGE	B_VHHAGE	B_VINCOME
Test 1	0.85	-3.85	0.03	-242.88	0	0	0	0	0	0	0	0	0	0	0	0
Test 2	0.86	-3.57	0.03	0.23	0	0	0	0	0	0	0	0	0	0	0	0
Test 3	0.81	-0.35	0.00	24.46	0	0	0	0	0	0	0	0	0	0	0	0
Test 4	0.83	-4.57	0.03	-265.03	0	0	0	0	0	0	0	0	0	0	0	0
Test 5	0.86	-11.41	0.08	-67.61	0	0	0	0	0	0	0	0	0	0	0	0
Test 6	0.82	-4.18	0.03	-3.24	0	0	0	0	0	0	0	0	0	0	0	0
Test 7	0.81	-3.33	0.03	1.67	0	0	0	0	0	0	0	0	0	0	0	0
Test 8	0.83	-6.40	0.04	-0.74	0	0	0	0	0	0	0	0	0	0	0	0

- iii) In our secondary regression, we regressed household inflection temperatures against income groups and other variables (type of residence [i.e., single-family home, multi-family home, condo, mobile home, townhouse], residence age, and residence size). However, we find that income is correlated with all three variables. Thus, because these variables are not independent of each other, including income, the type of residence, residence age, and residence size will introduce multicollinearity into the model, which interfere with our estimate of the relationship between income and inflection temperatures.
- iv) To further confirm this finding we use the variance inflation factor. In Table B (also added to Supplemental information Table S1 is the regression output with inflection temperature as the dependent variable, and type of residence, residence age, residence size as the independent variables, along with variance inflation factor for each independent variable. The variance inflation factor tells us if an independent variable is correlated with another variable in the same regression model. When the factor has a value of 1, the variable is not correlated with any other variable. The higher the variance inflation factor, the higher the chance of introducing multicollinearity into the model when both variables are included. We see that the residence size and income dummy variables have variance inflation factors close to or larger than 5¹⁷, meaning including them in the same model would be double counting their effects on the dependent variable, the inflection temperature. From these findings, we concluded that using income as the only dependent variable to be compared to the inflection temperature is appropriate. Nonetheless, despite the collinearity issue, Table B1 still shows that the highest income groups have the lowest inflection temperatures (as reflected by the negative coefficients), which is consistent with our main conclusion. In the main text we added a brief discussion of this independence challenge as follows (see line 522 in the main text): “We also considered the potential effects of type of residence [i.e., single-family home, multi-family home, condo, mobile home, townhouse], residence age, and residence size that can have on the household’s inflection temperature. However, we find including these variables would introduce multicollinearity into the model because they are correlated with income. For more details, please see Supplemental Information Table S1 and S2.”

¹⁷ How to Calculate Variance Inflation Factor (VIF) in R - Statology. <https://www.statology.org/variance-inflation-factor-r/>.

v) A deeper discussion of this is included in the Supplemental Information section, as follows:

“In our secondary regression, we regressed household inflection temperatures against income groups and other variables (type of residence [i.e., single-family home, multi-family home, condo, mobile home, townhouse], residence age, and residence size). However, we find that income is correlated with all three variables. Thus, because these variables are not independent of each other, including income, the type of residence, residence age, and residence size will introduce multicollinearity into the model, which interfere with our estimate of the relationship between income and inflection temperatures.

To further confirm this finding we use the variance inflation factor. In Table S1 is the regression output with inflection temperature as the dependent variable, and type of residence, residence age, residence size as the independent variables, along with variance inflation factor for each independent variable. The variance inflation factor tells us if an independent variable is correlated with another variable in the same regression model. When the factor has a value of 1, the variable is not correlated with any other variable. The higher the variance inflation factor, the higher the chance of introducing multicollinearity into the model when both variables are included. We see that the residence size and income dummy variables have variance inflation factors close to or larger than 5, meaning including them in the same model would be double counting their effects on the dependent variable, the inflection temperature. From these findings, we concluded that using income as the only dependent variable to be compared to the inflection temperature is appropriate. Nonetheless, despite the collinearity issue, Table S1 still shows that the highest income groups have the lowest inflection temperatures (as reflected by the negative coefficients), which is consistent with our main conclusion.”

Table B1. Linear regression output of different variables against household inflection temperature in 2015-2016. The intercept represents multi-family residence type, zero years of residence age and zero people in the household, residences less than 1000 sqft, and income less than \$15,000. VIF is the variance inflation factor.

Coefficients:	Estimate	Std. Error	t value	Pr(> t)	VIF
(Intercept)	66.31158	0.70212	94.445	< 2e-16	***

VRESTYPE_CONDO	1.8521	0.70242	2.637	0.0084	**	1.806
VRESTYPE_MOB_HOME	1.36428	0.68515	1.991	0.04653	*	1.892
VRESTYPE_SIN_FAM_HOM	-0.86064	0.52267	-1.647	0.09972	.	3.988
VRESTYPE_TOWNHOUSE	0.69792	0.66631	1.047	0.29497		2.113
VRESAGE_INT	0.08464	0.00638	13.266	< 2e-16	***	1.113
VHOUSEHOLD_INT	-0.56983	0.07039	-8.096	7.64E-16	***	1.136
VSQFEET_1000_1499	-0.07189	0.42889	-0.168	0.8669		3.59
VSQFEET_1500_1999	0.34525	0.43943	0.786	0.4321		4.522
VSQFEET_2000_2999	0.0882	0.4683	0.188	0.85062		4.394
VSQFEET_3000_3999	0.17167	0.5725	0.3	0.7643		2.334
VSQFEET_4000_PLUS	0.81655	0.75092	1.087	0.27693		1.642
VINCOME_15000_24999	-0.76377	0.59657	-1.28	0.20053		2.665
VINCOME_25000_34999	-0.15377	0.59376	-0.259	0.79566		2.784
VINCOME_35000_49999	-0.27569	0.5544	-0.497	0.61903		4.026
VINCOME_50000_74999	-0.70633	0.53743	-1.314	0.18883		5.291
VINCOME_75000_99999	-1.07265	0.55893	-1.919	0.05505	.	4.385
VINCOME_100000_149999	-1.74809	0.5604	-3.119	0.00183	**	4.71
VINCOME_150000_PLUS	-2.33734	0.59976	-3.897	9.91E-05	***	3.802

Signif. codes: 0 '***' 0.001 '**' 0.01 '*' 0.05 '.' 0.1 ' ' 1

- b) Income groups should be considered the household size.
- i) Because income and household size are both household-level factors recorded once at a moment in time, including them in the regression model did not change the model output (see Table A above).
- c) There are some sampling issues. It seems like each year has different households, so it is difficult to compare across the years. Fig. 4 should also show the summer and winter differences.
- i) Household changes across years are random and this is limited by the data supplied from the utility company. When evaluating energy equity at the population level, the ideal AC turn on point is set by the highest income group in any given year, because this group has the least stringent budget constraint. Another key factor is that the energy equity gap is designed to be a relative measure to gauge how a metropolitan area is progressing towards

energy equality. The relativity of our measure is designed to help utility companies and policy makers determine if populations are able to adapt equally to external factors.

- ii) When we limited our data to the 1,984 households that had complete 4 years of data, we find that the same pattern stands, as portrayed in the following graph (Figure A). In each boxplot of Figure A, n equals 1,984. We have added this Figure in Supplementary Information (Figure S14).

Figure A. Infection temperature calculated for households present in all four years of data, n = 1,984

- iii) We calculate the inflection temperature using one year’s worth of data and do not separate the winter and summer months. The inflection temperature is the minimum point on a household’s temperature response curve. In other words, it is the temperature at which the household spends the least amount of energy across an entire year. Arizona has such a high-heat climate leading to the majority of temperature-dependent energy usage going towards cooling rather than heating. Our current model also resembles the linear symmetric temperature response function used in building energy simulations¹⁸. For a region with more extreme winter and summer differences, a model separating out a heating turn on point and cooling turn on point would be a valuable future research initiative.

¹⁸ Fazeli, R., Ruth, M. & Davidsdottir, B. Temperature response functions for residential energy demand - A review of models. *Urban Climate* vol. 15 45–59 (2016).

d) LN 251 on page 10 and Table 1: I would like to know the reasons for having significant changes in retail electricity prices and cooling degree days in Years 3 and 4. Also, is Max temperature is from indoor or outdoor?

i) Thank you for pointing this out, we have clarified that changes in the energy equity gap occur one year after price and cooling degree changes and that in the table we have outdoor temperature (see below). The changes in retail price¹⁹ and cooling degree days²⁰ are calculated from data from official sources, the US Energy Information Administration and the National Oceanic and Atmospheric Administration (NOAA), respectively. The changes in retail prices are a result of the utility company changing what they charge the customer.

(1) We have clarified this as follows (see line 248 in the main text): “The energy equity gap narrowed by 1.2°F (-20.3%) between the first two years of our study, but then widened by 0.5°F (10.6%) and 2.3°F (44.2%) in the last three years of our study, as seen in Table 1. We find that a change in cooling degree days or residential electricity price corresponds to energy equity gap changes in the following year. Between years one and two, there was a 2.4 % increase in residential electricity prices and a 3.6% increase in cooling degree days. This corresponds to a 10.6% increase in the energy equity gap in year 3, caused mainly by low-income groups waiting longer to turn on their AC systems.”

ii) Cooling degree days are based on the assumption that when the outside temperature is at a certain threshold (e.g., 65°F), a household does not need cooling to be comfortable. Degree days are the difference between the daily temperature mean, (high temperature plus low temperature divided by two) and in our example 65°F. If the temperature mean is above 65°F, we subtract 65 from the mean and the result is Cooling Degree Days. When NOAA highlighted the changes in cooling degree days from year 3 to 4 this is a result of the changing climate. This could result from less heat waves in years 3 and 4 of our analysis when compared to years 1 and 2.

Table 1. Temperature, electricity price, and energy equity gap shifts in Arizona. Max Average Monthly Max Temperature is the average maximum temperature of the hottest month in a year.

¹⁹ U.S. Energy Information Administration (EIA) - Qb.

<https://www.eia.gov/opendata/qb.php?category=1012&sdid=ELEC.PRICE.AZ-RES.A>.

²⁰ Search | Climate Data Online (CDO) | National Climatic Data Center (NCDC). <https://www.ncdc.noaa.gov/cdo-web/search;jsessionid=5E9B713509CA7D789539AD530ABF62CC>.

Arizona Metric	Change from year to year		
	Year 1 to 2	Year 2 to 3	Year 3 to 4
Max Average Monthly Max Outdoor Temperature	1.0%	-0.2%	-0.2%
Cooling Degree Days (CDD)	3.6%	2.5%	-5.2%
Average residential electricity retail price (cents/kWh)	2.4%	2.7%	-2.7%
Energy Equity Gap	-20.3%	10.6%	44.2%

e) One of the most significant issues for this article’s measure is the lack of an indoor temperature setting.

i) The lack of thermostat data is a valid point, and we address this point in the limitations section. While the indoor temperature setting could provide more information on how people are using electricity and their set points, thermostat information would not reveal a consumer’s true temperature comfort preferences. The inability of thermostat data to identify true household preferences results from household occupants adjusting their thermostat based on multiple factors such as spending limits (i.e., disposable income), comfort, and energy conservation habits. Thus, a person might deliberately keep their thermostats at non-ideal temperatures (i.e., energy limiting behavior), rather than their true comfortable temperature, to save money or energy.

ii) We have clarified this limitation in our limitations section as follows (see line 401 in the main text): “One limitation of this study is a lack of indoor thermostat data and an inability to quantify the actual indoor temperature of homes. Thermostat data could provide more information about the willingness of occupants to consume energy for cooling and heating needs. However, thermostat information would not reveal a consumer’s true temperature comfort preferences. The inability of thermostat data to identify true household preferences results from household occupants adjusting their thermostat based on multiple factors such as spending limits (i.e., disposable income), comfort, and energy conservation habits. Thus, a person might deliberately keep their thermostats at non-ideal temperatures (i.e., energy limiting behavior), rather than their true comfortable temperature, to save money or energy.”

5) Energy insecurity comparison – it would be interesting to compare those households that forgo energy to meet other necessities vs. their counterparts.

a) We acknowledge that households experiencing poverty often forgo one service, like energy, to meet other basic needs^{21,22}. However, specific information on the basic needs households have

²¹ Herrero, S. T. Energy poverty indicators: A critical review of methods. *Indoor and Built Environment* **26**, 1018–1031 (2017).

chosen to forgo is not included within the survey provided by the electric utility company. In the paper we compare households that limit their energy consumption to those that spend large portions of their income on meeting their energy services. This could imply that households that spend a large portion of their income on their energy are forgoing other basic needs. In our analysis only 3 out of 4577 households in the first year are considered energy poor under both measures.

²² Sareen, S. *et al.* European energy poverty metrics: Scales, prospects and limits. *Global Transitions* **2**, 26–36 (2020).

Reviewer 2

This paper presents a novel approach to quantifying energy poverty by defining an energy equity gap and comparing the gap across income groups. The paper is well written with a thorough review of the literature, clear evidence and nicely presented methods. The authors make a strong case for the policy relevance of their approach by demonstrating how their metric identifies households that may forgo energy for other needs, which are not captured by the energy poverty metrics currently in use. That said, there are a few points in the paper that require clarification. So, I recommend the paper for publication with minor revisions. I list the points that need to be addressed below.

We thank the reviewer for the positive evaluation of the previous manuscript. Below we have addressed the reviewer's concerns in detail with additional analyses. Thanks for helping us improve the paper.

- 1) Line 65: county  country
 - a) Corrected, thank you for pointing this out!
- 2) Fig 1: A picky point about the placement of surveys on the 2x2 plot. As I understand it, survey-based data isn't necessarily relative. It can be used to determine whether a family is above or below an absolute threshold.
 - a) Thank you for bringing this to our attention. We have updated Figure 1 to clarify what surveys we were referring to, highlighting that surveys can fit in multiple categories (see below). We believe using survey outcomes to calculate a score is represented in the 4th quadrant (absolute-secondary), where allows a scoring threshold to be set for energy poverty. The second quadrant (primary-relative) refers to the types of surveys where only subjective indicators (e.g. are you able to cool your home to your desired level) and housing characteristics (e.g. do you feel your windows are drafty) are considered and no score is calculated. We have updated figure 1 to include the details above:

Figure 1. Categories of energy poverty metrics

- 3) Line 93: The authors should consider adding a footnote explaining the difference between equity and equality - some readers may not be familiar with the distinction.
 - a) We added the following to the text (See line 103 in the main text): *“Equality entails giving everyone the same tools to achieve a desired level of energy consumption. For example, having open access information on how low income households can apply for energy assistance on the web. Equity entails giving groups different types of tools such that they can equally take advantage of opportunities or reach a desired goal. An energy equity example would entail each person receiving enough assistance to cool or heat their homes to safe levels.”*
- 4) Line 94: is  are
 - a) Corrected, thank you for pointing this out!
- 5) Line 103: delete the 't'
 - a) Corrected, thank you for pointing this out!
- 6) Line 190-191: At this point I started to wonder whether the authors had any way to control for housing type (e.g. Multi-family vs. single), family size, or personal preference? The authors mention "thermostat preferences" on p. 13 and mention housing characteristics in their discussion of "Limitations", but perhaps it would be worth raising these issues earlier in the paper so that readers are aware that the authors have considered them.

- a) We added a sentence earlier on to refer people to the limitations section (see line 318 of the main text): “We present a more detailed discussion of preference limitation in the Supplemental Information notes.”
- b) Our regression model includes daily electricity prices based on the pricing plan each household uses. The prices may vary day to day for a household depending on the season, weekday or weekend, and whether it’s a holiday. We acknowledge that factors like housing characteristics (insulation, drafty windows, etc.) and household characteristics (income, number of people in a household, etc.) will affect how people use energy. However, our inflection temperature regression is run on individual households. Since these households factors are uniform across the entire time period for each household in our dataset, including them in the temperature response function model does not yield any effects, and these variables have coefficients of zero (See Table A below).
- c) When we included housing (heating type, number of ACs, AC type) and household characteristics (size, age, and members) we did not see an improvement in the model fit, evident in the zero coefficients of the added factors (see Table A below). Thus we exclude these factors from the inflection temperature regression model because household factors that are uniform across the modeling horizon will not affect model outputs. In the main text we highlight this point (Equation 1) as follows (see line 468 in the main text): “A household's inflection temperature is calculated using a nonlinear regression model (equation 1), which estimates daily electricity consumption of household i on day t ($E_{i,t}$) based on the following variables: daily average temperature (T_t), electricity price based on the billing plan of the household and season ($P_{i,s}$), dummy variables of whether day t is a holiday (H_t), day-of-the-week fixed effects (δ_t), and month-of-the-year fixed effects (μ_t). When modeling day-of-the-week and month-of-the-year dummy variables, Wednesday and March were dropped, respectively, to prevent collinearity.

$$E_{i,t} = \alpha + \beta_1 \times T_t + \beta_2 \times T_t^2 + \beta_3 \times P_{i,s} + H_t + \delta_t + \mu_t \quad (1)''$$

Table A. Coefficients of regression model for calculating inflection temperatures. The following factors were added to Equation 1 in the main text and did not change nor improve model fit: home heating type (VHOMEHEAT), home AC type (VACTYPE), number of AC units (VACUNITS), residence size (VSQFEET),

residence age(VRESAGE), age of the head of household(VHHAGE), the number of people in the household (VHOUSEHOLD_INT), and household income (VINCOME). All but residence size (VSQFEET), residence age(VRESAGE), age of the head of household(VHHAGE), the number of people in the household(VHOUSEHOLD_INT), and household income (VINCOME) are dummy variables. R squared and the coefficients of the original factors included in the model have not changed (coefficients of holidays, weekly and monthly fixed effects are not shown)

Household	r_squared	Coefficients														
		B_temp_avg	B_temp_avg_sq	B_elec_cost	B_VHOMEHEAT_ELEC	B_VACTYPE_GAS_HEAT	B_VACUNITS_TWO	B_VACUNITS_THREE_PLUS	B_VSQFEET_1000_1499	B_VSQFEET_1500_1999	B_VSQFEET_2000_2999	B_VSQFEET_3000_3999	B_VHOUSEHOLD_INT	B_VRESAGE	B_VHHAGE	B_VINCOME
Test 1	0.85	-3.85	0.03	-242.88	0	0	0	0	0	0	0	0	0	0	0	0
Test 2	0.86	-3.57	0.03	0.23	0	0	0	0	0	0	0	0	0	0	0	0
Test 3	0.81	-0.35	0.00	24.46	0	0	0	0	0	0	0	0	0	0	0	0
Test 4	0.83	-4.57	0.03	-265.03	0	0	0	0	0	0	0	0	0	0	0	0
Test 5	0.86	-11.41	0.08	-67.61	0	0	0	0	0	0	0	0	0	0	0	0
Test 6	0.82	-4.18	0.03	-3.24	0	0	0	0	0	0	0	0	0	0	0	0
Test 7	0.81	-3.33	0.03	1.67	0	0	0	0	0	0	0	0	0	0	0	0
Test 8	0.83	-6.40	0.04	-0.74	0	0	0	0	0	0	0	0	0	0	0	0

7) Line 236: The authors wrote "As the energy equity gap shrinks and widens, we see the driving factor is that low-income households are getting worse off while high-income households are improving". Causation implied by this sentence isn't totally clear. Are the authors saying that the widening EEG leaves low-income people worse off? It might be more accurate to say that a widening EEG exposes people to greater risk.

a) We have modified the sentence to (See line 240 in the main text): **"The energy equity gap shrinks and widens as low-income households experience higher inflection temperatures while high-income households lower their inflection temperatures, highlighting increasing energy inequity in our study region over time."**

- 8) Table 1: The largest change in EEG occurred between Year 3 and 4, but price and CDDs had both declined. Perhaps worth acknowledging this anomaly.
- a) We have clarified this as follows (see line 248 of the main text): “The energy equity gap narrowed by 1.2°F (-20.3%) between the first two years of our study, but then widened by 0.5°F (10.6%) and 2.3°F (44.2%) in the last three years of our study, as seen in Table 1. We find that a change in cooling degree days or residential electricity price corresponds to energy equity gap changes in the following year. Between years one and two, there was a 2.4 % increase in residential electricity prices and a 3.6% increase in cooling degree days. This corresponds to a 10.6% increase in the energy equity gap in year 3, caused mainly by low-income groups waiting longer to turn on their AC systems.”
- 9) Line 277: This page has the first mention of age. It appears to be an important covariate. However, the authors do not explain what age they use in the analysis. I assume it is the age of the household head, but it should be explained clearly.
- a) We have clarified in the text that this is the age of the head of the household (see line 285 in the main text): “In Figures 5c and 5d, we investigate energy poverty and equity across head of household age groups.”
- 10) Line 278: E-23 should be a number like 0.05?
- a) Corrected, thank you for pointing this out!
- 11) Lines 308-317: I appreciate that the authors attempted to address individual preferences. However, it isn't clear to me why preferences would cluster by ethnicity. Has this been shown in prior research? Should we also expect to see differences within ethnic groups?
- a) Prior studies found that genetic differences do not determine temperature preference^{23,24}, so by looking at Supplemental Information Figures S4-S7, we see the difference in inflection temperatures across ethnicities may be due to inequities, because we see a wide vertical distribution for the Black and Asian populations. If there was a preference difference between ethnicities, we would observe a narrower vertical distribution.

²³ Lawes, M., Havenith, G. & Hodder, S. Ethnic differences: The influence of relative humidity on thermal perception.

²⁴ Wang, L., Chen, M. & Yang, J. Interindividual differences of male college students in thermal preference in winter. *Building and Environment* **173**, 106744 (2020).

b) We have clarified this point in the text (see line 270 in the main text): “Comparing 5a and 5b and assuming no difference in temperature preference between ethnicities^{25,26}, we see that the overall inflection temperature is high in the Black population. Along with high energy equity gaps, meaning overall, the Black population is worse off and there is high inequity within the black population. In the Asian population, the overall median inflection temperatures are low yet there are wide energy equity gaps, indicating high income disparity within the group.”

12) In addition, in this section, the authors describe several patterns they observe between different ethnic groups. Specifically, they note how Black and Asian HHs have find a wide vertical distribution of inflection temperatures, while Latino HHs have a narrower distribution. When I visually inspect the Figs S4-6, I see that the widest distribution of Inflection Temp among Latino HHs (~10 deg F in 2018/19) is wider than the distribution among both Black and Asian HHs in some years. Are there statistical tests that the authors can use to show that the variation between groups they describe is significant? The SI skirts the issues. The authors state (SI Lines 78-79) "Black and Asian groups may be less tolerant of 78 heat, therefore have lower inflection temperatures overall " but they don't specify any statistical test. This leads me to believe that it isn't a significant relationship. If this it the case, the authors should note that the pattern exists, but is not significant and their discussion is speculative.

a) Thank you for pointing this out, according to your previous point, we have added our explanation on the difference we see across ethnicities. We have also conducted Mood's median tests and pairwise median tests on inflection temperatures by income, age, and ethnicity. We found that although ethnicity is the least correlated with inflection temperature compared to income and age, meaning the difference in inflection temperatures across ethnicities is less statistically significant than across income and age groups, there is still a pattern to be observed.

b) We conducted Pearson's Chi-squared tests between income and age, income and ethnicity, and age and ethnicity. The Pearson's Chi-squared test is used to quantify the correlation between two categorical variables, or in this case, the lack thereof. We found no correlation between each pair, meaning in each case, we fail to reject the null hypothesis that there is no correlation between each pairing of the variables. Since income, age, and ethnicity are not correlated with

²⁵ Lawes, M., Havenith, G. & Hodder, S. Ethnic differences: The influence of relative humidity on thermal perception.

²⁶ Wang, L., Chen, M. & Yang, J. Interindividual differences of male college students in thermal preference in winter. *Building and Environment* **173**, 106744 (2020).

one another, we can conclude the difference in inflection temperatures is more likely due to inequity rather than a difference in preference between ethnicities^{27,28}. Based on previous literature, the difference in inflection temperatures across age groups could be due to the ability to regulate body temperature for different age groups^{29,30,31,32,33,34}. We have included the following contingency tables and added the following text in the supplemental information: “To test the hypothesis that income, age of the head of household, and ethnicity are not correlated with each other, we conducted Pearson’s Chi-squared tests between income and age, income and ethnicity, and age and ethnicity. The Pearson’s Chi-squared test is used to quantify the correlation between two categorical variables. We found no correlation between each pair of variables (income-ethnicity, ethnicity-age, income-age), meaning in each case, we fail to reject the null hypothesis that there is no correlation between each pairing of the variables. Since income, age, and ethnicity are not correlated with one another, we can conclude the difference in inflection temperatures is more likely due to inequity rather than a difference in preference between ethnicities^{35,36}. Based on previous literature, the difference in inflection temperatures across age groups could be due to the ability to regulate body temperature for different age groups^{37,38,39,40,41,42}.”

²⁷ Lawes, M., Havenith, G. & Hodder, S. Ethnic differences: The influence of relative humidity on thermal perception.

²⁸ Wang, L., Chen, M. & Yang, J. Interindividual differences of male college students in thermal preference in winter. *Building and Environment* **173**, 106744 (2020).

²⁹ Frederick H. Rohles, JR. Preference for the Thermal Environment by the Elderly: <https://doi.org/10.1177/001872086901100106> **11**, 37–41 (2016).

³⁰ Li, P., Liu, Y. & Dong, J. Age-Related Thermal Comfort in a Science Museum with Hot–Humid Climate in Summer. *Environmental Science and Engineering* 421–431 (2019) doi:10.1007/978-981-13-9520-8_45.

³¹ Hwang, R.-L. & Chen, C.-P. Field study on behaviors and adaptation of elderly people and their thermal comfort requirements in residential environments. *Indoor Air* **20**, 235–245 (2010).

³² NA, T., NK, A. & DG, P. Preferred room temperature of young vs aged males: the influence of thermal sensation, thermal comfort, and affect. *The journals of gerontology. Series A, Biological sciences and medical sciences* **50**, (1995).

³³ M, N. *et al.* Regional differences in temperature sensation and thermal comfort in humans. *Journal of applied physiology (Bethesda, Md. : 1985)* **105**, 1897–1906 (2008).

³⁴ Natsume, K., Ogawa, T., Sugeno, J., Ohnishi, N. & Imai, K. Preferred ambient temperature for old and young men in summer and winter. *International Journal of Biometeorology* 1992 **36:1** **36**, 1–4 (1992).

³⁵ Lawes, M., Havenith, G. & Hodder, S. Ethnic differences: The influence of relative humidity on thermal perception.

³⁶ Wang, L., Chen, M. & Yang, J. Interindividual differences of male college students in thermal preference in winter. *Building and Environment* **173**, 106744 (2020).

³⁷ Frederick H. Rohles, JR. Preference for the Thermal Environment by the Elderly: <https://doi.org/10.1177/001872086901100106> **11**, 37–41 (2016).

³⁸ Li, P., Liu, Y. & Dong, J. Age-Related Thermal Comfort in a Science Museum with Hot–Humid Climate in Summer. *Environmental Science and Engineering* 421–431 (2019) doi:10.1007/978-981-13-9520-8_45.

Table C1. Contingency table of income and ethnicity in the dataset. X-squared = 196.27, df = 56, p-value < 2.2e-16

	White	Asian	Hispanic	Black	Other
Less than \$15,000	195	14	63	25	29
\$15,000 to \$24,999	372	19	78	20	22
\$25,000 to \$34,999	345	9	104	20	41
\$35,000 to \$49,999	587	29	201	38	44
\$50,000 to \$74,999	962	35	246	39	53
\$75,000 to \$99,999	638	42	144	25	43
\$100,000 to \$149,999	688	46	109	26	27
\$150,000 or more	440	38	46	13	25

Table C2. Contingency table of age of the head of household and ethnicity in the dataset. X-squared = 292.32, df = 56, p-value < 2.2e-16

	White	Asian	Hispanic	Black	Other
18-24 yrs old	95	12	36	9	14
25-34 yrs old	431	37	82	24	35
35-44 yrs old	759	81	326	58	75
45-54 yrs old	669	38	209	26	49
55-64 yrs old	782	26	136	34	45
65-74 yrs old	692	15	92	23	28
75+ yrs old	440	8	43	8	14

Table C3. Contingency table of income and age of the head of household in the dataset. X-squared = 412.76, df = 49, p-value < 2.2e-16

	18-24 yrs old	25-34 yrs old	35-44 yrs old	45-54 yrs old	55-64 yrs old	65-74 yrs old	75+ yrs old
Less than \$15,000	14	22	55	41	50	55	54
\$15,000 to \$24,999	18	34	61	58	84	115	103

³⁹ Hwang, R.-L. & Chen, C.-P. Field study on behaviors and adaptation of elderly people and their thermal comfort requirements in residential environments. *Indoor Air* **20**, 235–245 (2010).

⁴⁰ NA, T., NK, A. & DG, P. Preferred room temperature of young vs aged males: the influence of thermal sensation, thermal comfort, and affect. *The journals of gerontology. Series A, Biological sciences and medical sciences* **50**, (1995).

⁴¹ M, N. et al. Regional differences in temperature sensation and thermal comfort in humans. *Journal of applied physiology (Bethesda, Md. : 1985)* **105**, 1897–1906 (2008).

⁴² Natsume, K., Ogawa, T., Sugeno, J., Ohnishi, N. & Imai, K. Preferred ambient temperature for old and young men in summer and winter. *International Journal of Biometeorology* **36**, 1–4 (1992).

\$25,000 to \$34,999	15	44	102	77	88	96	65
\$35,000 to \$49,999	31	93	180	118	151	159	95
\$50,000 to \$74,999	44	162	291	208	224	190	102
\$75,000 to \$99,999	24	96	229	151	167	101	43
\$100,000 to \$149,999	10	106	244	189	149	90	33
\$150,000 or more	10	52	137	149	110	44	18

13) Lines 330-331: Here the authors discuss their selection of a "comfort set point" of 78F, above which HHs are defined as energy poor. For this they cite a study by Díaz et al. which found mortality risk increases above a daily max temp of 90F (actually they find risk really jumps at a max of 36.5F or ~98F). The authors should note that the study they cite refers to daily max temp, which is quite different than their comfort set point, which would theoretically be maintained for the entire day. Of course, their point of health-related illness vs. mortality still stands.

- a) To clarify, the 78 degrees Fahrenheit was chosen because it is the recommended indoor AC setting for government buildings^{43,44}, as well as many utility companies^{45,46}, to maintain a comfortable enough temperature while also being conscious of energy conservation. Depending on a household's insulation and how many days in a row a heat wave lasts, the indoor temperature may be lower or higher than the outdoor temperature.
- b) We have clarified this point in the text (See line 335 in the main text): *"We acknowledge that heat-mortality risk occurs when outdoor temperature rises above 90°F, as seen in Díaz et al (note their study of mortality risk used daily maximum temperatures). Our goal is to identify households at risk for both health-related illness and death, which can result from a lower temperature threshold. We derive this lower threshold (78°F) from recommended indoor AC setting for government buildings^{47,48}, as well as from recommendations of utility companies^{49,50}."*

⁴³ Energy Tips for Institutional and Government Buildings | ddoe. <https://doee.dc.gov/service/energy-tips-institutional-and-government-buildings>.

⁴⁴ Thermostats | Department of Energy. <https://www.energy.gov/energysaver/thermostats>.

⁴⁵ Thermostat Settings: The Ideal Settings for Summer & Winter. <https://valleyservice.net/blogs/thermostat-settings>.

⁴⁶ Recommended Thermostat Settings for Winter and Summer. <https://www.centralhtg.com/blog/recommended-thermostat-settings>.

⁴⁷ Energy Tips for Institutional and Government Buildings | ddoe. <https://doee.dc.gov/service/energy-tips-institutional-and-government-buildings>.

⁴⁸ Thermostats | Department of Energy. <https://www.energy.gov/energysaver/thermostats>.

⁴⁹ Thermostat Settings: The Ideal Settings for Summer & Winter. <https://valleyservice.net/blogs/thermostat-settings>.

⁵⁰ Recommended Thermostat Settings for Winter and Summer. <https://www.centralhtg.com/blog/recommended-thermostat-settings>.

14) Figure 7: Is the number at the top of each graph the mean, median, or some other metric. I did not see it specified anywhere.

a) We have clarified this on the graph and caption. It is the percent of households that spend more than 10% of their income on electricity. Please see below.

Figure 7. Energy poverty measured using the energy burden (EB) metric. The x-axis measures the percent of income a household spends on electricity. The red dotted line indicates the 10% income spending threshold. Most households spend between 1% and 10% of their income on electricity. EB₁₀ represents the proportion of households that spent over 10% of their income on electricity. For example, in 2015-2016, 2.7% of households in our sample spent more than 10% of their income on electricity.

15) Line 400: This is the only mention of SDGs in the entire document. Though they apply globally, the concept is generally used in the context of developing countries. I am not arguing that it doesn't apply to their analysis; however, none of the indicators for SDG7 specifically address the type of energy poverty that the authors present in this paper. If the authors do want to refer to the SDGs, I suggest they define SDG7 and add a short discussion in the intro explaining to readers why it's relevant for this context even if it this type of energy poverty isn't specifically addressed by the existing targets and indicators.

a) Thank you for pointing this out, we have removed any mentions of SDGs in the paper.

Reviewer 3

An interesting paper with some good analysis. There does seem to be some limitations which are mostly described in the Limitations section. Some detailed questions/comments:

Thank you for your thorough review and thoughtful suggestions, below are our responses and edits made according to your comments. Thank you for helping us improve our paper!

- 1) Line 65: “country-to-county” is this intended to have a second “r” to make it “country”
 - a) Corrected, thank you for pointing this out!
- 2) Line 76: “This metric does not assume...” Could you clarify which metric you are referring to – it seems to contrast to the prior sentence.
 - a) We have clarified this in the text (See line 77 in the main text): “Access-consumption matrices portray shifts in a country's energy profile, mainly the change in fuel utilization and how many people use each fuel. If more people are gaining access to energy services in an underdeveloped country, and more people are shifting from dirtier to cleaner fuels in developing regions, energy poverty is decreasing^{51,52}. Due to the fact that energy poverty in developing regions often means a lack of access to modern energy services, metrics measuring the level of access to modern energy services are best used for countries that are still building modern energy services and beginning their clean energy transition.”
- 3) Line 103: “t households “: typo?
 - a) Corrected, thank you for pointing this out!
- 4) Line 107-108: “energy poverty analysis in developed countries has been dominated by absolute-primary economic-based metrics” perhaps this is somewhat overstated as there are many studies on subjective indicators (some that you cited and many others)
 - a) We have changed the sentence to the following (See line 122 in the main text): “Despite the recent development of new metrics to capture consumer behaviors (e.g., under-consumption of energy and choice of thermal comfort)^{53,54,2,3}, household-level energy poverty evaluation for

⁵¹ Pachauri, S. & Spreng, D. Measuring and monitoring energy poverty. *Energy Policy* **39**, 7497–7504 (2011).

⁵² Culver, L. C. *Energy Poverty: What You Measure Matters*.

⁵³ Romero, J. C., Linares, P. & López, X. The policy implications of energy poverty indicators. *Energy Policy* **115**, 98–108 (2018).

⁵⁴ Herrero, S. T. Energy poverty indicators: A critical review of methods. *Indoor and Built Environment* **26**, 1018–1031 (2017).

government assistance programs in developed countries has been led by absolute-primary economic-based metrics^{55,56,57,58}.”

- 5) Line 114: “First, they are sensitive to energy prices and” is this a repeat of line 100?
 - a) We have deleted the first mention, thank you for the suggestion!
- 6) Line 123-124: “unsafe practices (e.g., using stove or space heaters to heat space” space heaters to heat space sounds like an intended activity which is not necessarily unsafe.
 - a) We have clarified this in the text (See line 138 within the main text): “Within developed countries (i.e., those with close to 100% electricity supply access)⁵⁹, energy poverty and insecurity can manifest themselves in 1) electricity shutoffs resulting from nonpayment, 2) forgoing heating services due to financial strain and participating in unsafe practices (e.g., using the stove or oven for heat, unsafe uses of space heating technologies which lead to fires⁶⁰), 3) spending a large percentage of income on energy bills, and 4) difficulty adopting clean energy and efficient technologies^{61,62,63,64}.”
- 7) Line 126-129: “we find a void of metrics that can identify energy-limiting households (i.e., those without comfortable indoor temperatures) who may put themselves at risk of heat-related illness, excess indoor moisture, mold growth, and other adverse health effects” Similar to the point for line 107-108, I think there are many papers on this.
 - a) We have added some more citations to the introduction and metrics discussion. Additionally, we have added the following to the text (See line 92 of the main text): “Although there are many

⁵⁵ Romero, J. C., Linares, P. & López, X. The policy implications of energy poverty indicators. *Energy Policy* **115**, 98–108 (2018).

⁵⁶ Bednar, D. J. & Reames, T. G. Recognition of and response to energy poverty in the United States. *Nature Energy* vol. 5 432–439 (2020).

⁵⁷ Liddell, C., Morris, C., McKenzie, S. J. P. & Rae, G. Measuring and monitoring fuel poverty in the UK: National and regional perspectives. *Energy Policy* **49**, 27–32 (2012).

⁵⁸ Moore, R. Definitions of fuel poverty: Implications for policy. *Energy Policy* **49**, 19–26 (2012).

⁵⁹ Access to electricity (% of population) | Data. <https://data.worldbank.org/indicator/EG.ELC.ACCS.ZS>.

⁶⁰ Common Causes of Electric Space Heater Fire Examined | J.S. Held.

<https://jsheld.com/insights/articles/common-causes-of-electric-space-heater-fires-methods-of-prevention>.

⁶¹ Reames, T. G. Targeting energy justice: Exploring spatial, racial/ethnic and socioeconomic disparities in urban residential heating energy efficiency. *Energy Policy* **97**, 549–558 (2016).

⁶² Reames, T. G. A community-based approach to low-income residential energy efficiency participation barriers. *Local Environment* **21**, 1449–1466 (2016).

⁶³ Hernández, D. & Siegel, E. Energy insecurity and its ill health effects: A community perspective on the energy-health nexus in New York City. *Energy Research and Social Science* **47**, 78–83 (2019).

⁶⁴ Memmott, T., Carley, S., Graff, M. & Konisky, D. M. Sociodemographic disparities in energy insecurity among low-income households before and during the COVID-19 pandemic. *Nature Energy* **6**, 186–193 (2021).

survey studies on housing characteristics and perceived energy poverty^{65,66,67}, we find a lack of metrics which can quantify energy limiting behavior. While surveys can elicit perceptions of energy limiting behavior, there is a need to have data driven approaches to determining the degree to which people actually limit their behavior.”

8) Line 148-149: “effect of weather on energy expenditure by examining trends in socioeconomic and ethnic groups.” The phrases may not fit together all that well.

a) We acknowledge that the term “weather” is misused here, we have changed this sentence to the following (See line 168 in the main text): “The energy equity gap metric considers the effect of outdoor temperature on energy consumption and quantifies relative energy limiting behavior, where those with fewer constraints on their budget set the threshold for a desired level of energy consumption to maintain a comfortable indoor temperature in the region. We examine the trends in energy limiting behavior across income, ethnic, and age groups.”

9) Line 150-151: “eliminates the effect of weather or temperature on electricity usage” outdoor temperature rather than indoor

a) We have clarified this in the text (See line 172 in the main text): “Since the energy equity gap measures electricity usage patterns between income groups within a region, it eliminates the effect of weather or outdoor temperature on electricity usage for different households, which might occur when the study area is too large.”

10) Line 155: “identify energy insecure households that are dangerously close to sinking into energy poverty” perhaps a reminder of how you are defining energy poverty could be useful

a) We have added the following to the beginning of the section (See line 148 in the main text): “We define energy poverty as having inadequate energy services within the household, or an inability to consume energy at a desired level. Thus a more holistic definition of energy poverty would include people who limit their energy consumption (i.e., display energy limiting behavior), and those who spend a large portion of their income on their energy bills (i.e., high energy burden).”

⁶⁵ Spiliotis, E., Arsenopoulos, A., Kanellou, E., Psarras, J. & Kontogiorgos, P. A multi-sourced data based framework for assisting utilities identify energy poor households: a case-study in Greece. <https://doi.org/10.1080/15567249.2020.1739783> **15**, 49–71 (2020).

⁶⁶ Middlemiss, L. *et al.* Energy poverty and social relations: A capabilities approach. *Energy Research and Social Science* **55**, 227–235 (2019).

⁶⁷ Memmott, T., Carley, S., Graff, M. & Konisky, D. M. Sociodemographic disparities in energy insecurity among low-income households before and during the COVID-19 pandemic. *Nature Energy* **2021 6:2 6**, 186–193 (2021).

11) Line 162: “assuming there is no difference in comfort preference or need across income groups” are these realistic assumptions and could there be large differences across groups or individuals? Do you control for age of household occupants in the main analysis? Perhaps older occupants have different income and also different heating preferences. Also for building characteristics, e.g. insulation.

a) We conducted Pearson’s Chi-squared tests between income and age, income and ethnicity, and age and ethnicity. The Pearson’s Chi-squared test is used to quantify the correlation between two categorical variables, or in this case, the lack thereof. We found no correlation between each pair, meaning in each case, we fail to reject the null hypothesis that there is no correlation between each pairing of the variables. Since income, age, and ethnicity are not correlated with one another, we can conclude the difference in inflection temperatures is more likely due to inequity rather than a difference in preference between ethnicities^{68,69}. Based on previous literature, the difference in inflection temperatures across age groups could be due to the ability to regulate body temperature for different age groups^{70,71,72,73,74,75}. We have included the following contingency tables and added the following text in the supplemental information: “To test the hypothesis that income, age of the head of household, and ethnicity are not correlated with each other, we conducted Pearson’s Chi-squared tests between income and age, income and ethnicity, and age and ethnicity. The Pearson’s Chi-squared test is used to quantify the correlation between two categorical variables. We found no correlation between each pair of variables (income-ethnicity, ethnicity-age, income-age), meaning in each case, we fail to reject the null hypothesis that there is no correlation between each pairing of the variables. Since

⁶⁸ Lawes, M., Havenith, G. & Hodder, S. Ethnic differences: The influence of relative humidity on thermal perception.

⁶⁹ Wang, L., Chen, M. & Yang, J. Interindividual differences of male college students in thermal preference in winter. *Building and Environment* **173**, 106744 (2020).

⁷⁰ Frederick H. Rohles, JR. Preference for the Thermal Environment by the Elderly: <https://doi.org/10.1177/001872086901100106> **11**, 37–41 (2016).

⁷¹ Li, P., Liu, Y. & Dong, J. Age-Related Thermal Comfort in a Science Museum with Hot–Humid Climate in Summer. *Environmental Science and Engineering* 421–431 (2019) doi:10.1007/978-981-13-9520-8_45.

⁷² Hwang, R.-L. & Chen, C.-P. Field study on behaviors and adaptation of elderly people and their thermal comfort requirements in residential environments. *Indoor Air* **20**, 235–245 (2010).

⁷³ NA, T., NK, A. & DG, P. Preferred room temperature of young vs aged males: the influence of thermal sensation, thermal comfort, and affect. *The journals of gerontology. Series A, Biological sciences and medical sciences* **50**, (1995).

⁷⁴ M, N. *et al.* Regional differences in temperature sensation and thermal comfort in humans. *Journal of applied physiology (Bethesda, Md. : 1985)* **105**, 1897–1906 (2008).

⁷⁵ Natsume, K., Ogawa, T., Sugeno, J., Ohnishi, N. & Imai, K. Preferred ambient temperature for old and young men in summer and winter. *International Journal of Biometeorology* 1992 36:1 **36**, 1–4 (1992).

income, age, and ethnicity are not correlated with one another, we can conclude the difference in inflection temperatures is more likely due to inequity rather than a difference in preference between ethnicities^{76,77}. Based on previous literature, the difference in inflection temperatures across age groups could be due to the ability to regulate body temperature for different age groups^{78,79,80,81,82,83}.”

Table C1. Contingency table of income and ethnicity in the dataset. X-squared = 196.27, df = 56, p-value < 2.2e-16

	White	Asian	Hispanic	Black	Other
Less than \$15,000	195	14	63	25	29
\$15,000 to \$24,999	372	19	78	20	22
\$25,000 to \$34,999	345	9	104	20	41
\$35,000 to \$49,999	587	29	201	38	44
\$50,000 to \$74,999	962	35	246	39	53
\$75,000 to \$99,999	638	42	144	25	43
\$100,000 to \$149,999	688	46	109	26	27
\$150,000 or more	440	38	46	13	25

Table C2. Contingency table of age of the head of household and ethnicity in the dataset. X-squared = 292.32, df = 56, p-value < 2.2e-16

⁷⁶ Lawes, M., Havenith, G. & Hodder, S. Ethnic differences: The influence of relative humidity on thermal perception.

⁷⁷ Wang, L., Chen, M. & Yang, J. Interindividual differences of male college students in thermal preference in winter. *Building and Environment* **173**, 106744 (2020).

⁷⁸ Frederick H. Rohles, JR. Preference for the Thermal Environment by the Elderly: <https://doi.org/10.1177/001872086901100106> **11**, 37–41 (2016).

⁷⁹ Li, P., Liu, Y. & Dong, J. Age-Related Thermal Comfort in a Science Museum with Hot–Humid Climate in Summer. *Environmental Science and Engineering* 421–431 (2019) doi:10.1007/978-981-13-9520-8_45.

⁸⁰ Hwang, R.-L. & Chen, C.-P. Field study on behaviors and adaptation of elderly people and their thermal comfort requirements in residential environments. *Indoor Air* **20**, 235–245 (2010).

⁸¹ NA, T., NK, A. & DG, P. Preferred room temperature of young vs aged males: the influence of thermal sensation, thermal comfort, and affect. *The journals of gerontology. Series A, Biological sciences and medical sciences* **50**, (1995).

⁸² M, N. *et al.* Regional differences in temperature sensation and thermal comfort in humans. *Journal of applied physiology (Bethesda, Md. : 1985)* **105**, 1897–1906 (2008).

⁸³ Natsume, K., Ogawa, T., Sugeno, J., Ohnishi, N. & Imai, K. Preferred ambient temperature for old and young men in summer and winter. *International Journal of Biometeorology* 1992 36:1 **36**, 1–4 (1992).

	White	Asian	Hispanic	Black	Other
18-24 yrs old	95	12	36	9	14
25-34 yrs old	431	37	82	24	35
35-44 yrs old	759	81	326	58	75
45-54 yrs old	669	38	209	26	49
55-64 yrs old	782	26	136	34	45
65-74 yrs old	692	15	92	23	28
75+ yrs old	440	8	43	8	14

Table C3. Contingency table of income and age of the head of household in the dataset. X-squared = 412.76, df = 49, p-value < 2.2e-16

	18-24 yrs old	25-34 yrs old	35-44 yrs old	45-54 yrs old	55-64 yrs old	65-74 yrs old	75+ yrs old
Less than \$15,000	14	22	55	41	50	55	54
\$15,000 to \$24,999	18	34	61	58	84	115	103
\$25,000 to \$34,999	15	44	102	77	88	96	65
\$35,000 to \$49,999	31	93	180	118	151	159	95
\$50,000 to \$74,999	44	162	291	208	224	190	102
\$75,000 to \$99,999	24	96	229	151	167	101	43
\$100,000 to \$149,999	10	106	244	189	149	90	33
\$150,000 or more	10	52	137	149	110	44	18

- b) Our regression model includes daily electricity prices based on the pricing plan each household uses. The prices may vary day to day for a household depending on the season, weekday or weekend, and whether it's a holiday. We acknowledge that factors like housing characteristics (insulation, drafty windows, etc.) and household characteristics (income, residence age, number of people in a household, etc.) will affect how people use energy. However, our inflection temperature regression is run on individual households. Since these households factors are uniform across the entire time period for each household in our dataset, including them in the temperature response function model does not yield any effects, and these variables have coefficients of zero (See Table A below, included in Supplemental Information Table S6).
- c) When we included housing (heating type, number of ACs, AC type) and household characteristics (size, age, and members) we did not see an improvement in the model fit, evident in the zero coefficients of the added factors (see Table A below). Thus we exclude these factors from the inflection temperature regression model because household factors that are

uniform across the modeling horizon will not affect model outputs. In the main text we highlight this point (Equation 1) as follows (see line 649 in the main text): “A household's inflection temperature is calculated using a nonlinear regression model (equation 1), which estimates daily electricity consumption of household i on day t ($E_{i,t}$) based on the following variables: daily average temperature (T_t), electricity price based on the billing plan of the household and season ($P_{i,s}$), dummy variables of whether day t is a holiday (H_t), day-of-the-week fixed effects (δ_t), and month-of-the-year fixed effects (μ_t). When modeling day-of-the-week and month-of-the-year dummy variables, Wednesday and March were dropped, respectively, to prevent collinearity.

$$E_{i,t} = \alpha + \beta_1 \times T_t + \beta_2 \times T_t^2 + \beta_3 \times P_{i,s} + H_t + \delta_t + \mu_t \quad (1)''$$

Table A. Coefficients of regression model for calculating inflection temperatures. The following factors were added to Equation 1 in the main text and did not change nor improve model fit: home heating type (VHOMEHEAT), home AC type (VACTYPE), number of AC units (VACUNITS), residence size (VSQFEET), residence age (VRESAGE), age of the head of household (VHHAGE), the number of people in the household (VHOUSEHOLD_INT), and household income (VINCOME). All but residence size (VSQFEET), residence age (VRESAGE), age of the head of household (VHHAGE), the number of people in the household (VHOUSEHOLD_INT), and household income (VINCOME) are dummy variables. R squared and the coefficients of the original factors included in the model have not changed (coefficients of holidays, weekly and monthly fixed effects are not shown)

Household	Coefficients															
	r_squared	B_temp_avg	B_temp_avg_sq	B_elec_cost	B_VHOMEHEAT_ELEC	B_VACTYPE_GAS_HEAT	B_VACUNITS_TWO	B_VACUNITS_THREE_PLUS	B_VSQFEET_1000_1499	B_VSQFEET_1500_1999	B_VSQFEET_2000_2999	B_VSQFEET_3000_3999	B_VHOUSEHOLD_INT	B_VRESAGE	B_VHHAGE	B_VINCOME
Test 1	0.85	-3.85	0.03	-242.88	0	0	0	0	0	0	0	0	0	0	0	0
Test 2	0.86	-3.57	0.03	0.23	0	0	0	0	0	0	0	0	0	0	0	0

Test 3	0.81	-0.35	0.00	24.46	0	0	0	0	0	0	0	0	0	0	0	0
Test 4	0.83	-4.57	0.03	-265.03	0	0	0	0	0	0	0	0	0	0	0	0
Test 5	0.86	-11.41	0.08	-67.61	0	0	0	0	0	0	0	0	0	0	0	0
Test 6	0.82	-4.18	0.03	-3.24	0	0	0	0	0	0	0	0	0	0	0	0
Test 7	0.81	-3.33	0.03	1.67	0	0	0	0	0	0	0	0	0	0	0	0
Test 8	0.83	-6.40	0.04	-0.74	0	0	0	0	0	0	0	0	0	0	0	0

- d) In our secondary regression, we regressed household inflection temperatures against income groups and other variables (type of residence [i.e., single-family home, multi-family home, condo, mobile home, townhouse], residence age, and residence size). However, we find that income is correlated with all three variables. Thus, because these variables are not independent of each other, including income, the type of residence, residence age, and residence size will introduce multicollinearity into the model, which interfere with our estimate of the relationship between income and inflection temperatures. Also we cannot include specific information about insulation in the paper due to this not being included in the survey. We believe the household age provides insight into the insulation question.
- e) To further confirm this finding we use the variance inflation factor. In Table B (also added to Supplemental information Table S1 is the regression output with inflection temperature as the dependent variable, and type of residence, residence age, residence size as the independent variables, along with variance inflation factor for each independent variable. The variance inflation factor tells us if an independent variable is correlated with another variable in the same regression model. When the factor has a value of 1, the variable is not correlated with any other variable. The higher the variance inflation factor, the higher the chance of introducing multicollinearity into the model when both variables are included. We see that the residence size and income dummy variables have variance inflation factors close to or larger than 5⁸⁴, meaning including them in the same model would be double counting their effects on the dependent variable, the inflection temperature. From these findings, we concluded that using income as the only dependent variable to be compared to the inflection temperature is appropriate. Nonetheless, despite the collinearity issue, Table B1 still shows that the highest income groups have the lowest inflection temperatures (as reflected by the negative coefficients), which is consistent with our main conclusion. In the main text we added a brief

⁸⁴ How to Calculate Variance Inflation Factor (VIF) in R - Statology. <https://www.statology.org/variance-inflation-factor-r/>.

discussion of this independence challenge as follows (see line 520 in the main text): “We also considered the potential effects of type of residence [i.e., single-family home, multi-family home, condo, mobile home, townhouse], residence age, and residence size that can have on the household’s inflection temperature. However, we find including these variables would introduce multicollinearity into the model because they are correlated with income. For more details, please see Supplemental Information Table S1 and S2.”

- f) A deeper discussion of the secondary regression is included in the Supplemental Information section, as follows: “In our secondary regression, we regressed household inflection temperatures against income groups and other variables (type of residence [i.e., single-family home, multi-family home, condo, mobile home, townhouse], residence age, and residence size). However, we find that income is correlated with all three variables. Thus, because these variables are not independent of each other, including income, the type of residence, residence age, and residence size will introduce multicollinearity into the model, which interfere with our estimate of the relationship between income and inflection temperatures.

To further confirm this finding we use the variance inflation factor. In Table S1 is the regression output with inflection temperature as the dependent variable, and type of residence, residence age, residence size as the independent variables, along with variance inflation factor for each independent variable. The variance inflation factor tells us if an independent variable is correlated with another variable in the same regression model. When the factor has a value of 1, the variable is not correlated with any other variable. The higher the variance inflation factor, the higher the chance of introducing multicollinearity into the model when both variables are included. We see that the residence size and income dummy variables have variance inflation factors close to or larger than 5, meaning including them in the same model would be double counting their effects on the dependent variable, the inflection temperature. From these findings, we concluded that using income as the only dependent variable to be compared to the inflection temperature is appropriate. Nonetheless, despite the collinearity issue, Table S1 still shows that the highest income groups have the lowest inflection temperatures (as reflected by the negative coefficients), which is consistent with our main conclusion.”

Table B1. Linear regression output of different variables against household inflection temperature in 2015-2016. The intercept represents multi-family residence type, zero years of residence age and zero people in the household, residences less than 1000 sqft, and income less than \$15,000. VIF is the variance inflation factor.

Coefficients:	Estimate	Std. Error	t value	Pr(> t)		VIF
(Intercept)	66.31158	0.70212	94.445	< 2e-16	***	
VRESTYPE_CONDO	1.8521	0.70242	2.637	0.0084	**	1.806
VRESTYPE_MOB_HOME	1.36428	0.68515	1.991	0.04653	*	1.892
VRESTYPE_SIN_FAM_HOM	-0.86064	0.52267	-1.647	0.09972	.	3.988
VRESTYPE_TOWNHOUSE	0.69792	0.66631	1.047	0.29497		2.113
VRESAGE_INT	0.08464	0.00638	13.266	< 2e-16	***	1.113
VHOUSEHOLD_INT	-0.56983	0.07039	-8.096	7.64E-16	***	1.136
VSQFEET_1000_1499	-0.07189	0.42889	-0.168	0.8669		3.59
VSQFEET_1500_1999	0.34525	0.43943	0.786	0.4321		4.522
VSQFEET_2000_2999	0.0882	0.4683	0.188	0.85062		4.394
VSQFEET_3000_3999	0.17167	0.5725	0.3	0.7643		2.334
VSQFEET_4000_PLUS	0.81655	0.75092	1.087	0.27693		1.642
VINCOME_15000_24999	-0.76377	0.59657	-1.28	0.20053		2.665
VINCOME_25000_34999	-0.15377	0.59376	-0.259	0.79566		2.784
VINCOME_35000_49999	-0.27569	0.5544	-0.497	0.61903		4.026
VINCOME_50000_74999	-0.70633	0.53743	-1.314	0.18883		5.291
VINCOME_75000_99999	-1.07265	0.55893	-1.919	0.05505	.	4.385
VINCOME_100000_149999	-1.74809	0.5604	-3.119	0.00183	**	4.71
VINCOME_150000_PLUS	-2.33734	0.59976	-3.897	9.91E-05	***	3.802

Signif. codes: 0 '***' 0.001 '**' 0.01 '*' 0.05 '.' 0.1 ' ' 1

12) Line 195: “until later in the summer” how do you know it’s later in the summer – are the temperatures consistently hotter in late summer without much variation between days?

- a) The outdoor temperature peaks in July in Arizona, but daily high temperatures can reach 95 degrees Fahrenheit in May. Temperatures in Arizona consistently increase from January to July, and consistently decrease from August to December⁸⁵.
- b) We have included the following date-temperature graph in the Supplemental Information Figure S15:

Figure B. Monthly average temperatures in the study area, data taken from usclimatedata.com⁸⁶

- 13) Line 235-236: “As the energy equity gap shrinks and widens, we see the driving factor is that low-income households are getting worse off while high-income households are improving.” Perhaps there is scope to improve this sentence. Is it focusing on changes over time?
- a) We have modified the sentence to (See line 240 in the main text): “The energy equity gap shrinks and widens as low-income households experience higher inflection temperatures while high-income households lower their inflection temperatures, highlighting increasing energy inequity in the region of study over time.”
- 14) Line 244-245: “This may be caused by the delayed price elasticity of demand effects in year-to-year residential electricity price changes and a warming climate” Can it also just be random variation from the comparison of point estimates?

⁸⁵ Climate Phoenix - Arizona and Weather averages Phoenix.
<https://www.usclimatedata.com/climate/phoenix/arizona/united-states/usaz0166>.

⁸⁶ Climate Phoenix - Arizona and Weather averages Phoenix.
<https://www.usclimatedata.com/climate/phoenix/arizona/united-states/usaz0166>.

- a) There is a chance that this could result from random variation in the point estimates of the population mean. In a general sense if we can only choose one value to estimate the population mean (our mid-point), and then take a new sample of 100 people and recompute the mean; there is a chance that we will not get the exact same. However, we computed the energy equity gap using a subset of our population that had data consistent across all four years of our study and found similar trends. Thus we believe this trend is not solely due to variation.
- i) 1,984 households that had complete 4 years of data, we find that the same pattern stands, as portrayed in the following graph (Figure A). In each boxplot of Figure A, n equals 1,984.

Figure A. Infection temperature calculated for households present in all four years of data, n = 1,984

- 15) Line 260-317: is the sample size large enough for this analysis? There are some large changes between years in Figure 5. Are all the variables e.g. age, income, ethnicity included as controls in a single estimation at any point of the paper?
- a) We use the largest sample size permitted based on the data provided by the utility company, totaling 6002 households across 4 years. Please see the following contingency tables. We conducted Pearson’s Chi-squared tests between income and age, income and ethnicity, and age and ethnicity. The Pearson’s Chi-squared test is used to quantify the correlation between two categorical variables, or in this case, the lack thereof. We found no correlation between each pair, meaning in each case, we fail to reject the null hypothesis that there is no correlation

between each pairing of the variables, so we did not include all three variables in a single estimation. Since income, age, and ethnicity are not correlated with one another, we can conclude the difference in inflection temperatures is more likely due to inequity rather than a difference in preference between ethnicities^{7,8}. Based on previous literature, the difference in inflection temperatures across age groups could be due to the ability to regulate body temperature for different age groups^{87,88,89,90,91,92}. We have included the following contingency tables and added the following text in the supplemental information: “To test the hypothesis that income, age of the head of household, and ethnicity are not correlated with each other, we conducted Pearson’s Chi-squared tests between income and age, income and ethnicity, and age and ethnicity. The Pearson’s Chi-squared test is used to quantify the correlation between two categorical variables. We found no correlation between each pair of variables (income-ethnicity, ethnicity-age, income-age), meaning in each case, we fail to reject the null hypothesis that there is no correlation between each pairing of the variables. Since income, age, and ethnicity are not correlated with one another, we can conclude the difference in inflection temperatures is more likely due to inequity rather than a difference in preference between ethnicities^{93,94}. Based on previous literature, the difference in inflection temperatures across age groups could be due to the ability to regulate body temperature for different age groups.”

⁸⁷ Frederick H. Rohles, JR. Preference for the Thermal Environment by the Elderly:

<https://doi.org/10.1177/001872086901100106> **11**, 37–41 (2016).

⁸⁸ Li, P., Liu, Y. & Dong, J. Age-Related Thermal Comfort in a Science Museum with Hot–Humid Climate in Summer. *Environmental Science and Engineering* 421–431 (2019) doi:10.1007/978-981-13-9520-8_45.

⁸⁹ Hwang, R.-L. & Chen, C.-P. Field study on behaviors and adaptation of elderly people and their thermal comfort requirements in residential environments. *Indoor Air* **20**, 235–245 (2010).

⁹⁰ NA, T., NK, A. & DG, P. Preferred room temperature of young vs aged males: the influence of thermal sensation, thermal comfort, and affect. *The journals of gerontology. Series A, Biological sciences and medical sciences* **50**, (1995).

⁹¹ M, N. *et al.* Regional differences in temperature sensation and thermal comfort in humans. *Journal of applied physiology (Bethesda, Md. : 1985)* **105**, 1897–1906 (2008).

⁹² Natsume, K., Ogawa, T., Sugeno, J., Ohnishi, N. & Imai, K. Preferred ambient temperature for old and young men in summer and winter. *International Journal of Biometeorology* 1992 **36**, 1–4 (1992).

⁹³ Lawes, M., Havenith, G. & Hodder, S. Ethnic differences: The influence of relative humidity on thermal perception.

⁹⁴ Lawes, M., Havenith, G. & Hodder, S. Ethnic differences: The influence of relative humidity on thermal perception.

Table C1. Contingency table of income and ethnicity in the dataset. X-squared = 196.27, df = 56, p-value < 2.2e-16

	White	Asian	Hispanic	Black	Other
Less than \$15,000	195	14	63	25	29
\$15,000 to \$24,999	372	19	78	20	22
\$25,000 to \$34,999	345	9	104	20	41
\$35,000 to \$49,999	587	29	201	38	44
\$50,000 to \$74,999	962	35	246	39	53
\$75,000 to \$99,999	638	42	144	25	43
\$100,000 to \$149,999	688	46	109	26	27
\$150,000 or more	440	38	46	13	25

Table C2. Contingency table of age of the head of household and ethnicity in the dataset. X-squared = 292.32, df = 56, p-value < 2.2e-16

	White	Asian	Hispanic	Black	Other
18-24 yrs old	95	12	36	9	14
25-34 yrs old	431	37	82	24	35
35-44 yrs old	759	81	326	58	75
45-54 yrs old	669	38	209	26	49
55-64 yrs old	782	26	136	34	45
65-74 yrs old	692	15	92	23	28
75+ yrs old	440	8	43	8	14

Table C3. Contingency table of income and age of the head of household in the dataset. X-squared = 412.76, df = 49, p-value < 2.2e-16

	18-24 yrs old	25-34 yrs old	35-44 yrs old	45-54 yrs old	55-64 yrs old	65-74 yrs old	75+ yrs old
Less than \$15,000	14	22	55	41	50	55	54
\$15,000 to \$24,999	18	34	61	58	84	115	103
\$25,000 to \$34,999	15	44	102	77	88	96	65
\$35,000 to \$49,999	31	93	180	118	151	159	95
\$50,000 to \$74,999	44	162	291	208	224	190	102
\$75,000 to \$99,999	24	96	229	151	167	101	43
\$100,000 to \$149,999	10	106	244	189	149	90	33
\$150,000 or more	10	52	137	149	110	44	18

16) Line 361: “dangerously high inflection temperatures” does this depend on characteristics of the occupants? E.g. some high temperatures may not be a big deal for young and healthy occupants

a) We have added the following discussion to the main text (see line 324 of the main text): “We acknowledge that there are multiple factors that can influence the risk of occupants in high heat temperatures. The inflection temperature and the threshold temperature we choose to be the energy poverty cutoff serves as a first screening for identifying households that are at higher risk of putting themselves in danger of overheating.”

b) To address the different characteristics of the occupants, we investigate how the inflection temperature varies across ethnicities and age groups. Prior studies found that genetic differences do not determine temperature preference^{7,8}, so by looking at Supplemental Information Figures S4-S7, we see the difference in inflection temperatures across ethnicities may be due to inequities, because we see a wide vertical distribution for the Black and Asian populations. If there was a preference difference between ethnicities, we would observe a narrower vertical distribution, as we see across age groups in Supplemental Information Figures S8-S13.

17) Line 422: “identifying those falling through the policy cracks by casting a finer net.” Is it finer or just different?

a) We clarify by adding this sentence (see line 442 in the main text): “We can start by identifying those falling through the policy cracks of economic-based poverty metrics by casting a finer net which also includes poverty displayed through energy consumption behavior.”

18) Line 500: Does Table 3 mean that all groups are different to each other e.g. all 8 income groups are different to all of the others? Are there more details on the stars? Are the income, age, and ethnicity variables used in each part of the analysis, or are they included just separately?

a) From the further analyses outlined in point15 above, we decided to include income, ethnicity, and age separately.

b) We have added the significance codes to the table as follows (see line 527 in the main text):

Table 3. P-values from t Mood’s Median tests on median inflection temperatures of income, ethnicity, and age groups.

Grouping	2015-2016	2016-2017	2017-2018	2018-2019
Income	5.36E-23***	6.85E-22***	2.50E-16***	2.05E-13***
Ethnicity	5.94E-03*	6.02E-05***	2.85E-01	1.73E-02.
Age	2.30E-29***	1.61E-32***	8.91E-20***	7.36E-17***

*Signif. codes: 0 '***' 0.001 '**' 0.01 '*' 0.05 '.' 0.1 ' ' 1*

c) We have changed the Kruskal-Wallis test to the Mood’s Median test, because the latter is more robust against samples with outliers. We also conducted post-hoc pairwise median tests to see how each income/age/ethnicity group compare with each other. For income groups, although some neighboring groups are not statistically different from each other, two to three groups over indicate strong significance that the income groups have statistically different sample medians, where lower income groups have generally higher inflection temperatures, and higher income groups have lower inflection temperatures. Neighboring age groups are generally statistically significant from each other, besides the 18-24 and 35-44 groups. However, the general trend of younger groups having lower inflection temperatures and older groups having higher inflection temperatures still applies. There is the most uncertainty in the difference between ethnicities, which could be due to the limited data points for minorities, reflective of the study region. We have included this discussion in the Supplemental Information.

Table D1. P-value matrix of pairwise median test of inflection temperatures across income groups in 2015-2016. Red indicates values > 0.05.

	Less than \$15,000 to \$24,999	\$15,000 to \$24,999	\$25,000 to \$34,999	\$35,000 to \$49,999	\$50,000 to \$74,999	\$75,000 to \$99,999	\$100,000 to \$149,999
\$15,000 to \$24,999	5.71E-02						
\$25,000 to \$34,999	5.71E-02	9.45E-01					
\$35,000 to \$49,999	1.53E-02	2.98E-01	2.55E-01				
\$50,000 to \$74,999	1.70E-04	6.19E-02	2.82E-02	1.55E-01			
\$75,000 to \$99,999	6.20E-06	1.17E-02	4.02E-03	2.01E-02	2.21E-01		
\$100,000 to \$149,999	5.89E-10	3.69E-06	5.43E-08	7.14E-09	1.36E-06	4.05E-04	
\$150,000 or more	2.43E-12	2.27E-09	2.43E-12	3.91E-12	4.88E-10	2.20E-07	1.53E-02

Table D2. P-value matrix of pairwise median test of inflection temperatures across age groups in 2015-2016. Red indicates values > 0.05.

	18-24	25-34	35-44	45-54	55-64	65-74
25-34	7.96E-01					
35-44	6.69E-01	1.67E-02				
45-54	6.69E-01	3.37E-02	7.97E-01			
55-64	3.59E-02	1.25E-02	3.29E-09	3.36E-09		

65-74	8.13E-03	2.65E-04	6.75E-11	5.34E-11	1.74E-02	
75+	2.19E-05	3.16E-13	1.18E-20	7.19E-21	3.48E-10	3.18E-05

Table D3. P-value matrix of pairwise median test of inflection temperatures across ethnicity groups in 2015-2016. Red indicates values > 0.05.

	White	Asian	Hispanic
Asian	0.27		
Hispanic	0.04945	0.4585	
Black	1	0.27	0.5808

Reviewer comments, second round

Reviewer #1 (Remarks to the Author):

The Energy equity gap: Unveiling hidden energy poverty.

I appreciate the authors' efforts in improving the paper. However, although this revision has improved, I still find some fundamental issues.

Energy poverty definition and measures are still an issue. I don't think outdoor temperature alone is a good indication for energy poverty unless we know indoor temperature setting, energy consumption and bills, and building environment. On the other hand, it seems like inflection temperature is related to the difference between indoor and outdoor temperatures? Can the authors make it even more apparent in the beginning? The results become clear, but the introduction is not so clear. Additionally, the definition of energy poverty is still clear enough regarding the specific energy poverty measure in the following sentence, "having inadequate energy services within the household, or an inability to consume energy at the desired level, thus a more holistic definition of energy poverty would include people who limit their energy consumption (i.e., display energy limiting behavior), and those who spend a large portion of their income on their energy bills (i.e., high energy burden)."

I agree with the authors that energy burden alone is not an energy poverty indicator, although this is the most critical factor. It is a good idea to include behavior-based energy poverty measures as reasonable measures; however, the specific measures of energy behavioral part are not inclusive enough in this paper, such as energy consumption (or saving) behaviors, appliance usage and ownership, weatherization, etc. Additionally, it is unclear on the so-called energy limited behavior.

More importantly, one of the critical factors missing from the analysis is the renter or owner status, dwelling characteristics, appliance ownership, and seasonal effects as control variables. Still, not including them due to the multicollinearity is not a good reason. The analysis should consist of the interaction effect between income and age (age is generally a confounding factor in much demographic analysis).

I don't quite agree with the statement, "While the survey can elicit perceptions of energy limited behavior, there is a need to have data-driven approaches to determining the degree to which people limited the behavior. " Survey is one of the best ways to measure the majority of people's experience on energy poverty because it's not possible to observe energy insecurity situations in daily life directly, especially on low-income households. Additionally, the big data-driven approaches often cannot reach those underserved communities (older people with the internet) because low-income or vulnerable populations lack technology (e.g., smart meters, smart thermostats, and internet access during the data gathering process relating to energy security or poverty).

There are a few measurement questions that are not clear, and I listed a few below

- How is the eligibility of LIHEAP and WAP measured?
- It's unclear if the data analysis included household sociodemographic information, dwelling characteristics, and appliance ownership since they are listed in the method section. If they are not included, it should be pointed out earlier.
- Is the Time of Use program included in the analysis?
- Why Wednesday and March were dropped? Why do they have collinearity? The authors should deal with all the collinearity issues instead of dropping the variables and, for example, examining different races in a different model, different models for different seasons, etc.

Reviewer #2 (Remarks to the Author):

I am satisfied with the authors' responses to the concerns I raised about their original submission. I also think that they responded to the numerous valid concerns raised by other reviewers and recommend publishing with no additional revisions.

Reviewer #3 (Remarks to the Author):

Just some minor points:

Missing full stop on line 12

Line 17: "our of 4,577 households"

Line 183: "This is most likely occurs"

Line 251: "a delayed price elasticity of demand effects"

Line 391: "Approxzimately"

Line 437-9: "high-income households with high inflection temperatures may be best suited for discounted weatherization programs due to their likelihood of having limited disposable income."

Should this be low-income?

Is there text missing on line 647?

Energy Equity Gap: Unveiling hidden energy poverty

Response to reviewer comments, second revision*

***Reviewer comments are indicated in black, our responses in blue, and our text additions in red.**

References are at the end of each review

Reviewer 1

We thank the reviewer for their comments, below are our responses and edits.

I appreciate the authors' efforts in improving the paper. However, although this revision has improved, I still find some fundamental issues.

Energy poverty definition and measures are still an issue. I don't think outdoor temperature alone is a good indication for energy poverty unless we know indoor temperature setting, energy consumption and bills, and building environment. On the other hand, it seems like inflection temperature is related to the difference between indoor and outdoor temperatures? Can the authors make it even more apparent in the beginning? The results become clear, but the introduction is not so clear.

We agree with the reviewer's point that outdoor temperature alone is not enough, which is why we include energy consumption and electricity price in our model of the household-level temperature response functions. Our metric is not just purely based on outdoor temperature, but also based on energy consumption. We then combine our metric with bill/expenditure information to have a more comprehensive assessment of energy poverty. Please see equation 1 below, where we estimate daily electricity consumption of household i on day t ($E_{i,t}$) based on the following variables: daily average temperature (T_t), electricity price based on the billing plan of the household and season ($P_{i,s}$), dummy variables of whether day t is a holiday (H_t), day-of-the-week fixed effects (δ_t), and month-of-the-year fixed effects (μ_t). When modeling day-of-the-week and month-of-the-year dummy variables, Wednesday and March were dropped, respectively, to prevent collinearity. To address the point on indoor temperature setting, in reality, most utility companies and policymakers do not have indoor temperature information, so our metric is based on the available information the majority policymakers and agencies could have, making the metric easily accessible for policymakers to utilize when designing equitable energy policies.

$$E_{i,t} = \alpha + \beta_1 \times T_t + \beta_2 \times T_t^2 + \beta_3 \times P_{i,s} + H_t + \delta_t + \mu_t \quad (1)$$

In regard to the relationship between the inflection temperature and the indoor/outdoor temperature difference, without more granular data, we are not able to determine whether there is a relationship between the two. However, we speculate that there would not be a significant relationship between the two because the inflection temperature only represents the AC turning-on point, the indoor/outdoor temperature difference would instead affect the slope of the temperature response functions or the absolute amount of energy a household uses for indoor temperature control. We address the lack of indoor temperature readings in the Limitations section. Studies have also shown that there is a positive correlation between indoor and outdoor temperature, where indoor temperatures are dependent on outdoor temperatures and building characteristics when cooling and heating appliances are not in use¹⁻³. Since our study area, Arizona, experiences hot and dry summers, factors like humidity and other air conditions have less influence on indoor temperatures than other areas where summers are more humid. When adapting the energy equity gap to other regions, additional weather and climate factors may be considered as necessary to provide a more accurate representation of indoor temperature based on available outdoor temperatures.

Although beyond the scope of this paper, we are investigating the relationship between inflection temperatures and the slope of the temperature response function. The slope or steepness of the function tells us the amount of energy a household is consuming compared to others at any given temperature. For example, for two households with the same inflection temperature and similar base loads (non-temperature-dependent energy use), but one has a much steeper slope, it means this household is consuming more energy than the other household at every temperature. The extra energy could be from the household having more appliances or lower energy efficiency and worse insulation. Preliminary results show there is no distinct relationship between inflection temperatures and slopes, meaning having a low or high inflection temperature does not necessarily correlate with the amount of energy a household spends.

Additionally, the definition of energy poverty is still clear enough regarding the specific energy poverty measure in the following sentence, “having inadequate energy services within the household, or an inability to consume energy at the desired level, thus a more holistic definition of energy poverty would include people who limit their energy consumption (i.e., display energy

limiting behavior), and those who spend a large portion of their income on their energy bills (i.e., high energy burden).”

We have added a clarifying sentence to the definition of “energy limiting behavior” in the main text (line 176) as follows: “We define energy limiting behavior as a household’s inability or unwillingness to consume enough energy to reach a desired level of comfort. A household displays energy limiting behavior if they reduce their energy consumption significantly below another household within the same region that does not have a budget constraint for energy spending. For example, assume households A and B live in the same region and have similar preferences for their ideal indoor temperature, around 70°F. Household A is a low-income household (i.e., a budget constraint on energy spending), and Household B is a high-income household (i.e., no budget constraint). If Household B starts using their air conditioning unit when it is 70°F outside, but Household A waits until it is 75°F outside, then household A is displaying 5°F of energy limiting behavior compared to Household B.”

I agree with the authors that energy burden alone is not an energy poverty indicator, although this is the most critical factor. It is a good idea to include behavior-based energy poverty measures as reasonable measures; however, the specific measures of energy behavioral part are not inclusive enough in this paper, such as energy consumption (or saving) behaviors, appliance usage and ownership, weatherization, etc.

We agree that energy burden is a valuable metric for energy poverty identification. We would like to clarify that we are not arguing that the energy equity gap should replace energy burden as the primary energy poverty indicator. Instead, we believe that the two should be used in conjunction to capture multiple types of energy poverty that households could experience. Up to this point, there has not been a systematic way to identify households that limit their energy usage, or experience an energy deficit. Figure 1 below is a visual representation of the households that the energy burden metric misses but are captured by the energy equity gap.

Also, energy consumption behavior is directly included in our model. The regression function, seen in Equation 1 below, determines the inflection temperature based on daily energy consumption. We have included appliance ownership (number of AC units a household has) in the secondary regression (Supplemental Information Note 8: Regression between inflection

temperatures, housing characteristics and demographics), and found no statistically significant relationship between the number of AC units and inflection temperature. We do not have direct information about weatherization (survey limitation); but have included a discussion on residence age (an indirect measure of original weatherization standards). However, when adapting the energy equity gap in the future, if weatherization information is available, we agree that it would provide a more accurate estimate of indoor temperatures. We have added more to the discussion in the limitations section.

Figure 1. Venn diagram representation of the number of households captured by the energy equity gap (EEG) vs. energy burden (EB), 2015-2016. EEG 1st tier is energy insecurity, EEG 2nd tier is energy poverty. Only three households overlap between EEG 2nd tier (energy poor) and those with an energy burden greater than 10%.

$$E_{i,t} = \alpha + \beta_1 \times T_t + \beta_2 \times T_t^2 + \beta_3 \times P_{i,s} + H_t + \delta_t + \mu_t \quad (1)$$

Equation 1 details: we estimate daily electricity consumption of household i on day t ($E_{i,t}$) based on the following variables: daily average temperature (T_t), electricity price based on the billing plan of

the household and season ($P_{i,s}$), dummy variables of whether day t is a holiday (H_t), day-of-the-week fixed effects (δ_t), and month-of-the-year fixed effects (μ_t). When modeling day-of-the-week and month-of-the-year dummy variables, Wednesday and March were dropped, respectively, to prevent collinearity.

Additionally, it is unclear on the so-called energy limited behavior.

We have clarified the definition of energy limiting behavior as follows (line 176 in the main text): “We define energy limiting behavior as a household’s inability or unwillingness to consume enough energy to reach a desired level of comfort. A household displays energy limiting behavior if they reduce their energy consumption significantly below another household within the same region that does not have a budget constraint for energy spending. For example, assume households A and B live in the same region and have similar preferences for their ideal indoor temperature, around 70°F. Household A is a low-income household (i.e., a budget constraint on energy spending), and Household B is a high-income household (i.e., no budget constraint). If Household B starts using their air conditioning unit when it is 70°F outside, but Household A waits until it is 75°F outside, then household A is displaying 5°F of energy limiting behavior compared to Household B.”

More importantly, one of the critical factors missing from the analysis is the renter or owner status, dwelling characteristics, appliance ownership, and seasonal effects as control variables. Still, not including them due to the multicollinearity is not a good reason. The analysis should consist of the interaction effect between income and age (age is generally a confounding factor in much demographic analysis).

We have included seasonal effects in our model as monthly effects. Monthly effects (12 segments) capture the seasonal variation at a higher temporal resolution than four season designations. This higher resolution improves the fit of our regression model. We have included appliance ownership (number of AC units a household has, the major temperature-dependent appliance) in this analysis. Results show that the number of AC units does not affect inflection temperature (table 3), but the number of AC units are correlated with income (table 2). We regressed residence age with income and found the two to be highly correlated with each other (p-value < 2.2e-16), so we do not investigate residence age separately with inflection temperatures. Our dataset does not include renter/owner information, however even if we know if a household

rents or not, we will still need to know if they pay for electricity out of pocket or if their bill is included in the set monthly rent. We agree more information on this would contribute to future analysis.

We respectfully disagree with the collinearity statement. Having predictors that are highly correlated with each other can cause the following problems⁴: 1) the estimated coefficients can change wildly based on what other predictors are included in the model, 2) multicollinearity reduces the precision of the estimated coefficients and weakens the power of the proposed model. According to the textbook “Understanding Regression Analysis”, chapter 37 “The Problem of Multicollinearity”⁵, multicollinearity or collinearity occurs when two or more independent variables are highly correlated with each other.

To avoid undermining the significance of an independent variable, in this case, income level, we need to first identify any correlations between independent variables (income vs. residence age, size, and type) before performing a regression analysis on our target dependent variable (inflection temperature). We found that residence age, residence size, and residence type are all highly correlated with income (table 2). When we include the same factors to predict inflection temperatures, we find the effect of income weakens significantly (table 3). We also include ethnicity and age in the regression against income and found a correlation, although the coefficients are small for ethnicities and age, meaning different ethnicities have a slight difference in income group distribution (about half an income level), and increasing one year in age lowered a household by 0.03 income levels. Therefore, we maintain our initial analysis where we evaluate inflection temperatures by income, ethnicity, and age separately.

We have performed a separate regression analysis (please see Supplemental Information Note 10) including housing characteristics, specifically residence age, appliance ownership, type of residence, and residence size into the temperature response function, and found that they did not influence household-level inflection temperatures for the reason that they are household specific, and uniform across all data points for each particular household. When including the same factors in the regression with inflection temperature as the dependent variable, we found that these variables would introduce multicollinearity into the model because they are correlated with income.

Table 2. Linear regression output of different variables against household income group. The intercept represents multi-family residence type, zero years of residence age and zero people in the household, residences less than 1000 sqft, and with one AC unit. CONDO, MOBILE_HOME, SINGLE_FAMILY_HOME, and TOWNHOUSE are the other types of residences; RESIDENCE_AGE_INTEGER is the age of the building in integer years; HOUSEHOLD_SIZE_INTEGER is the number of people in the household; SIZE_SQFEET variables are the brackets for the size of residence in square feet; ACUNITS are the number of AC units a household has; ETHNIC variables are ethnicities, AGE_INT is the integer age of the head of household.

Coefficients:	Estimate	Std. Error	t value	Pr(> t)	
(Intercept)	5.431692	0.198677	27.339	<2e-16	***
CONDO	-0.01358	0.194364	-0.07	0.944299	
MOBILE_HOME	-0.71196	0.198194	-3.592	0.000332	***
SINGLE_FAMILY_HOME	0.262174	0.146813	1.786	0.074222	.
TOWNHOUSE	0.274148	0.184649	1.485	0.137711	
RESIDENCE_AGE_INTEGER	-0.00414	0.001766	-2.346	0.019037	*
HOUSEHOLD_SIZE_INTEGER	-0.01057	0.021225	-0.498	0.618596	
SIZE_SQFEET_1000_1499	0.771301	0.118426	6.513	8.40E-11	***
SIZE_SQFEET_1500_1999	1.342806	0.119969	11.193	<2e-16	***
SIZE_SQFEET_2000_2999	1.93462	0.131847	14.673	<2e-16	***
SIZE_SQFEET_3000_3999	2.204995	0.170755	12.913	<2e-16	***
SIZE_SQFEET_4000_PLUS	2.568061	0.233134	11.015	<2e-16	***
ACUNITS_TWO	0.482717	0.085949	5.616	2.10E-08	***
ACUNITS_THREE_PLUS	0.768275	0.197409	3.892	0.000101	***
ETHNIC_ASIAN	-0.1631	0.142514	-1.144	0.252515	
ETHNIC_BLACKA	-0.32426	0.186556	-1.738	0.08227	.
ETHNIC_HISP	-0.55496	0.07692	-7.215	6.57E-13	***
ETHNIC_OTHER	-0.4591	0.168757	-2.72	0.00655	**
AGE_INT	-0.03106	0.001986	-15.64	<2e-16	***

Signif. codes: 0 '***' 0.001 '**' 0.01 '*' 0.05 '.' 0.1 '' 1

Table 3. Linear regression output of different variables against household inflection temperature in 2015-2016. The intercept represents multi-family residence type, zero years of residence age and zero people in the household, residences less than 1000 sqft, income less than \$15,000, and with one AC unit. VIF is the variance inflation factor. CONDO, MOBILE_HOME, SINGLE_FAMILY_HOME, and TOWNHOUSE are the other types of residences; RESIDENCE_AGE_INTEGER is the age of the building in integer years; HOUSEHOLD_SIZE_INTEGER is the number of people in the household; SIZE_SQFEET variables are the brackets for the size of residence in square feet; INCOME variables are income brackets; ACUNITS are the number of AC units a household has.

Coefficients:	Estimate	Std. Error	t value	Pr(> t)		VIF
(Intercept)	66.36236	0.739378	89.754	< 2e-16	***	
CONDO	1.694445	0.712611	2.378	0.01747	*	1.87
MOBILE_HOME	1.474602	0.721836	2.043	0.04114	*	1.86
SINGLE_FAMILY_HOME	-0.78091	0.535774	-1.458	0.14506		4.06
TOWNHOUSE	0.732204	0.674062	1.086	0.27744		2.20
RESIDENCE_AGE_INTEGER	0.085387	0.006404	13.333	< 2e-16	***	1.11
HOUSEHOLD_SIZE_INTEGER	-0.56163	0.070076	-8.015	1.48E-15	***	1.13
SIZE_SQFEET_1000_1499	-0.20266	0.438831	-0.462	0.64424		3.79
SIZE_SQFEET_1500_1999	0.251433	0.447885	0.561	0.57457		4.74
SIZE_SQFEET_2000_2999	-0.01388	0.495039	-0.028	0.97763		4.98
SIZE_SQFEET_3000_3999	0.133145	0.637963	0.209	0.83469		2.95
SIZE_SQFEET_4000_PLUS	0.419606	0.87177	0.481	0.63031		2.29
INCOME_15000_24999	-0.64764	0.618048	-1.048	0.29476		2.76
INCOME_25000_34999	-0.18295	0.609811	-0.3	0.76418		2.97
INCOME_35000_49999	-0.53293	0.572424	-0.931	0.35192		4.31
INCOME_50000_74999	-0.946	0.55598	-1.701	0.08894	.	5.71
INCOME_75000_99999	-1.16608	0.575186	-2.027	0.0427	*	4.74
INCOME_100000_149999	-1.8975	0.577287	-3.287	0.00102	**	5.07
INCOME_150000_PLUS	-2.504	0.614387	-4.076	4.69E-05	***	4.10
ACUNITS_TWO	0.021326	0.315746	0.068	0.94615		1.88
ACUNITS_THREE_PLUS	0.662092	0.721987	0.917	0.35918		1.67

Signif. codes: 0 '***' 0.001 '**' 0.01 '*' 0.05 '.' 0.1 ' ' 1

I don't quite agree with the statement, "While the survey can elicit perceptions of energy limited behavior, there is a need to have data-driven approaches to determining the degree to which people limited the behavior. " Survey is one of the best ways to measure the majority of people's experience on energy poverty because it's not possible to observe energy insecurity situations in daily life directly, especially on low-income households. Additionally, the big data-driven approaches often cannot reach those underserved communities (older people with the internet) because low-income or vulnerable populations lack technology (e.g., smart meters, smart thermostats, and internet access during the data gathering process relating to energy security or poverty.

While we do not deny that surveys are a valuable tool, data driven approaches to identifying energy poor households will provide a needed complement to traditional surveys to identify energy poor households. With the wide spread of smart meters in residential homes, currently 75% of all US households covered⁶⁻⁸, and 100% of households covered in our dataset (including 23% low-income households (income < \$35,000) and 16% elderly households (age > 65)), low-income households will be able to participate in these data analyses. **To the reviewer's point**, surveys are now often conducted online and through survey agencies that provide participants compensation for completing surveys. These factors would likely skew survey demographics towards tech-savvy younger populations, and also making surveys less reliable because people may be completing them for the money incentive and providing careless information. It is much more difficult for smart meters to be tampered with, and provide utilities with scalable, reliable, and timely data for policy decisions. Smart meters also run on a separate network, so a household does not necessarily need to know how to use the internet or Wi-Fi to be connected via a smart meter network⁹.

We have altered the phrasing of the main text as follows (line 107): "There is a need to use surveys in combination with data driven approaches to elicit perceptions of energy limited behavior, while also determining the degree to which people actually limit energy consumption."

There are a few measurement questions that are not clear, and I listed a few below

- How is the eligibility of LIHEAP and WAP measured?

Both LIHEAP and WAP use income limit by household size to determine eligibility^{10,11}, with some flexibility for states to determine what income level to set as the eligibility threshold (e.g. \$32,348 for a household of 2 in Arizona). We have updated this in the main text of the paper.

- It's unclear if the data analysis included household sociodemographic information, dwelling characteristics, and appliance ownership since they are listed in the method section. If they are not included, it should be pointed out earlier.

We have performed separate analyses on inflection temperatures vs income, ethnicity, and age, and included this in our discussion, as well as adding dwelling characteristic distributions to the supplemental information (please see Supplemental Information Note 8). We have removed the mention of appliance ownership from the main text.

- Is the Time of Use program included in the analysis?

Yes, we include time of use pricing in the analysis. Our electric utility collaborator provided us with each household's pricing plan, and the details of available pricing plans from the Salt River Project. The electricity price is included in our inflection temperature calculation. Additionally, for each pricing plan, we calculated a weighted average price per kWh for each season detailed in the pricing plans. Total household electricity expenditures were calculated by multiplying daily consumption and seasonal unit prices. We have clarified this in the methods.

- Why Wednesday and March were dropped? Why do they have collinearity? The authors should deal with all the collinearity issues instead of dropping the variables and, for example, examining different races in a different model, different models for different seasons, etc.

In this case, we deal with the collinearity issue by removing variables, in line with the standard textbooks^{4,5}. When working with non-continuous or discrete time series data, we need to drop one month out of a year or one day out of a week, because knowing eleven out of twelve months or six out of seven days will tell us whether a day is in the twelfth month or on the seventh day, these are often referred to as dummy variables. Wednesday and March were randomly picked to be dropped, as it does not matter with day or month to drop, as long as one is dropped. We do examine different ethnicities and ages separately, please refer to Figure 5 in the main text.

References

1. Zurbier, M. *et al.* Street temperature and building characteristics as determinants of indoor heat exposure. *The Science of the total environment* **766**, (2021).
2. White-Newsome, J. L. *et al.* Climate change and health: indoor heat exposure in vulnerable populations. *Environmental research* **112**, 20–27 (2012).

3. Nguyen, J. L., Schwartz, J. & Dockery, D. W. The relationship between indoor and outdoor temperature, apparent temperature, relative humidity, and absolute humidity. *Indoor air* **24**, 103–112 (2014).
4. Multicollinearity in Regression Analysis: Problems, Detection, and Solutions - Statistics By Jim. <https://statisticsbyjim.com/regression/multicollinearity-in-regression-analysis/>.
5. The problem of multicollinearity. *Understanding Regression Analysis* 176–180 (1997) doi:10.1007/978-0-585-25657-3_37.
6. 75% of US households have smart meters – report. <https://www.smart-energy.com/industry-sectors/smart-meters/75-of-us-households-have-smart-meters-report/>.
7. • U.S. cumulative smart meter installations 2021 | Statista. <https://www.statista.com/statistics/676472/number-of-smart-meter-installations-in-the-united-states/>.
8. Cooper, A. & Shuster, M. Electric Company Smart Meter Deployments: Foundation for a Smart Grid (2021 Update).
9. Smart Meters and Wi-Fi | Smart Energy GB. <https://www.smartenergygb.org/about-smart-meters/mythbusting-smart-meter-problems/smart-meter-and-wifi>.
10. Low Income Home Energy Assistance Program (LIHEAP) | Benefits.gov. <https://www.benefits.gov/benefit/623>.
11. How to Apply for Weatherization Assistance | Department of Energy. <https://www.energy.gov/eere/wap/how-apply-weatherization-assistance>.

Reviewer 2

I am satisfied with the authors' responses to the concerns I raised about their original submission. I also think that they responded to the numerous valid concerns raised by other reviewers and recommend publishing with no additional revisions.

Thank you for your review and confidence in our study!

Reviewer 3

Just some minor points:

Missing full stop on line 12

Line 17: "our of 4,577 households"

Line 183: "This is most likely occurs"

Line 251: "a delayed price elasticity of demand effects"

Line 391: "Approxzimately"

Thank you so much for pointing these out, we have addressed them in the main text.

Line 437-9: "high-income households with high inflection temperatures may be best suited for discounted weatherization programs due to their likelihood of having limited disposable income."

Should this be low-income?

It is high-income, because weatherization costs may still be high for high-income households with limited disposable income. They would be receiving the benefit of weatherization instead of direct financial help. We have clarified this in the main text: "high-income households with high inflection temperatures may be best suited for discounted weatherization programs. Despite having higher incomes, weatherization costs may still be too high for these households if they have limited disposal income."

Is there text missing on line 647?

We don't believe there is text missing, but have done a through grammar check, and added text in other parts of the manuscript.

Reviewer comments, third round

Reviewer #3 (Remarks to the Author):

looks good